# Causal Imitation Learning under Expert-Observable and Expert-Unobservable Confounding

**Daqian Shao**[*]
University of Oxford
Oxford, UK
daqian.shao@cs.ox.ac.uk

**Thomas Kleine Buening**
ETH Zurich
Zurich, Switzerland

**Marta Kwiatkowska**
University of Oxford
Oxford, UK

## Abstract

We propose a general framework for causal Imitation Learning (IL) with hidden confounders, which subsumes several existing settings. Our framework accounts for two types of hidden confounders: (a) variables observed by the expert but not by the imitator, and (b) confounding noise hidden from both. By leveraging trajectory histories as instruments, we reformulate causal IL in our framework into a Conditional Moment Restriction (CMR) problem. We propose DML-IL, an algorithm that solves this CMR problem via instrumental variable regression, and upper bound its imitation gap. Empirical evaluation on continuous state-action environments, including Mujoco tasks, demonstrates that DML-IL outperforms existing causal IL baselines.

## 1 Introduction

Imitation Learning (IL) has emerged as a prominent paradigm in machine learning, where the objective is to learn a policy that mimics the behaviour of an expert by learning from their demonstrations. While classical IL theory suggests that, with infinite data, the IL error should vanish (Ross et al., 2011), practical implementations often yield suboptimal and unsafe behaviours (Lecun et al., 2005; Kuefler et al., 2017; Bansal et al., 2018). Prior work attributes these failures to various factors, including spurious correlations (de Haan et al., 2019; Codevilla et al., 2019; Pfrommer et al., 2023), temporal noise (Swamy et al., 2022b), expert-exclusive knowledge (Choudhury et al., 2017; Chen et al., 2019; Swamy et al., 2022a; Vuorio et al., 2022) and causal delusions (Ortega & Braun, 2008; Ortega et al., 2021), which act as *confounding variables* unobserved by the imitator. Previous work typically addresses these factors in isolation. In practice, however, these challenges can coexist, making partial solutions insufficient. This calls for a holistic approach that accounts for multiple confounding factors simultaneously.

We propose a general framework for causal imitation learning that models hidden confounders, i.e., variables present in the environment but not recorded in demonstrations. Importantly, we distinguish between *expert-observable* confounders, which influence expert decisions but are not accessible to the imitator, and *expert-unobservable* confounders, which introduce spurious correlations and remain hidden from both the imitator and the expert. As a result, our framework generalises prior settings and enables a broader, more realistic problem formulation. In previous work, it has been shown that the application of an interactive IL algorithm such as DAgger (Ross et al., 2011), which allows us to directly query the expert, can be effective in dealing with hidden confounders. However, an interactive expert is not a realistic assumption in many domains and applications. Therefore, we aim to develop approaches that solely rely on a fixed set of demonstrations.

Specifically, we propose an IL method that leverages trajectory histories as *Instrumental Variables* (IVs) to mitigate spurious correlations caused by expert-unobservable confounders. Additionally, by learning a history-dependent policy, we can infer information about expert-observable confounders, which enables us to better imitate the expert despite lacking access to said variables. We show that

---

[*]Alternative address: shaodaqian@gmail.com

IL in our framework can be reformulated as a *Conditional Moment Restriction* (CMR) problem—a well-studied problem in econometrics and causal inference, which allows us to design practical algorithms with theoretical guarantees on the imitation gap.

In summary, our main contributions are as follows:

- We introduce a framework for causal IL (Section 3) that incorporates both expert-observable and expert-unobservable confounding variables to unify and generalise many of the settings in previous work (e.g., Swamy et al. (2022b;a); Ortega et al. (2021); Vuorio et al. (2022)).

- We reformulate the problem of confounded IL in our framework as solving a CMR problem, where we aim to learn a history-dependent policy by leveraging trajectory histories as instruments to break the confounding (Section 4).

- We propose DML-IL, a novel IL algorithm in our framework, for which we prove an upper bound on the imitation gap that recovers prior works' results as special cases (Theorem 4.5).

- We empirically validate our algorithm in both custom and MuJoCo environments with both expert-observable and expert-unobservable confounders and demonstrate that DML-IL outperforms existing causal IL baselines (Section 5). This highlights the need to explicitly account for both types of hidden confounders.

## 1.1 RELATED WORKS

**Causal Imitation Learning.** Imitation learning considers the problem of learning from demonstrations (Pomerleau, 1988; Lecun et al., 2005). Standard IL methods include Behaviour Cloning (Pomerleau, 1988), inverse RL (Russell, 1998), and adversarial methods (Ho & Ermon, 2016). Interactive IL (Ross et al., 2011) extends standard IL by allowing the imitator to query an interactive expert, facilitating recovery from mistakes. However, in this paper, we do not assume query access to an interactive expert. Recently, it has been shown that IL from offline trajectories can suffer from the existence of latent variables (Ortega et al., 2021; Bica et al., 2021), which cause causal delusion. This can be resolved by learning an interventional policy. Following this discovery, various methods (Vuorio et al., 2022; Swamy et al., 2022a) consider IL when the expert has access to the full hidden context that is fixed throughout each episode, whereas the imitator does not observe the hidden context. They aim to learn an interventional policy through on-policy IL algorithms that require an interactive demonstrator and/or an interactive simulator (e.g., DAgger (Ross et al., 2011)).

Orthogonal to these works, Swamy et al. (2022b) consider latent variables unobserved by the expert, which act as confounding noise that affects the recorded expert demonstrations, but not the transition dynamics. To address this challenge, the problem is then cast into an IV regression problem. Our work combines and generalises the above works (Vuorio et al., 2022; Swamy et al., 2022a;b) to allow the latent variables to be (a) only partly known to the expert, (b) evolving through time in each episode, and (c) directly affecting both the expert policy and the transition dynamics. Solving this generalisation implies solving the above problems simultaneously.

Causal confusion (de Haan et al., 2019; Pfrommer et al., 2023) considers the situation where the expert's actions are spuriously correlated with non-causal features of the previous observable states. While it is implicitly assumed that there are no latent variables present in the environment, we can still model this spurious correlation as the existence of hidden confounders that affect both previous states and current expert actions. Slight variations of this setting have been studied in Wen et al. (2020); Spencer et al. (2021); Codevilla et al. (2019). In Appendix A, we explain and discuss how these works can be reduced to special cases of our general framework. From the perspective of causal inference (Kumor et al., 2021; Zhang et al., 2020), previous work has studied the theoretical conditions on the causal graph under which the imitator can exactly match the expert performance through backdoor adjustments (*imitability*). Hereto related, Ruan et al. (2023) extended such conditions and backdoor adjustments to inverse RL. We instead consider a setting where exact imitation is impossible and aim to minimise the imitation gap. Beyond backdoor adjustments, imitability has also been studied theoretically using context-specific independence relations (Jamshidi et al., 2023). Finally, Ruan et al. (2024) analyse IL under unobserved confounding and show that exact imitation is impossible without additional assumptions. They develop robust IL algorithms tailored to such partially identifiable regimes. In contrast, we adopt structural assumptions (finite-horizon and additive confounding noise) which induce a valid instrumental-variables relation in the trajectory history.

These stronger assumptions avoid their impossibility result and yield point identification of the history-dependent policy, although the expert's latent variables themselves remain unidentifiable.

**IV Regression and Conditional Moment Restrictions (CMRs).** In this paper, we transform the causal IL problem into solving a CMR problem through IVs, to which end we provide a brief overview over IV regression and approaches for solving CMRs. The classic IV regression algorithms mainly consider linear functions (Angrist et al., 1996) and non-linear basis functions (Newey & Powell, 2003; Chen & Christensen, 2018; Singh et al., 2019). More recently, deep neural networks have been used for function approximation and methods such as DeepIV (Hartford et al., 2017), DeepGMM (Bennett et al., 2019b), AGMM (Dikkala et al., 2020), DFIV (Xu et al., 2020) and DML-IV (Shao et al., 2024) have been proposed. More generally, IV regression algorithms can be generalised to solve CMRs (Liao et al., 2020; Dikkala et al., 2020; Shao et al., 2024), specifically linear CMRs, where the restrictions are linear functionals of the function of interest. In our paper, we derive linear CMRs for causal IL so that the above methods can be adopted.

## 2 PRELIMINARIES: IVS AND CMRS

We first introduce the concept of Instrumental Variables (IVs) and its connection to Conditional Moment Restrictions (CMRs). Consider a structural model for outcome $Y$ and treatment $X$:

$$Y = f(X) + \varepsilon(U) \ \text{ with } \ \mathbb{E}[\varepsilon(U)] = 0, \tag{1}$$

where $U$ is a hidden confounder that affects both $X$ and $Y$ so that $\mathbb{E}[\varepsilon(U) \mid X] \neq 0$. Due to the presence of this hidden confounder, standard regressions (e.g., ordinary least squares) generally fail to produce consistent estimates of the causal relationship between $X$ on $Y$, i.e., $f(X)$. If we only have observational data, a classic technique for learning $f$ is IV regression (Newey & Powell, 2003). An IV $Z$ is an observable variable that satisfies the following conditions:

- *Unconfounded Instrument*: $Z \perp\!\!\!\perp U$;
- *Relevance*: $\mathbb{P}(X|Z)$ is not constant in $Z$;
- *Exclusion*: $Z$ does not directly affect $Y$: $Z \perp\!\!\!\perp Y \mid (X, U)$.

Using IVs, we are able to formulate the problem of learning $f$ into a CMR problem (Dikkala et al., 2020), where we aim to solve for $f$ satisfying $\mathbb{E}[Y - f(X) \mid Z] = 0$. In our work, we show that we can use trajectory histories as instruments to learn the causal relationship between states and expert actions by transforming the problem of causal IL into a CMR problem (Section 4).

## 3 A GENERAL CAUSAL IMITATION LEARNING FRAMEWORK

**MDPs with Hidden Confounders.** We now introduce a general framework for causal IL in the presence of hidden confounders. We begin by introducing a Markov Decision Process (MDP) formulation with hidden confounders, $(\mathcal{S}, \mathcal{A}, \mathcal{U}, \mathcal{P}, r, \mu_0, T)$, where $\mathcal{S}$ is the state space, $\mathcal{A}$ is the action space and $\mathcal{U}$ is the confounder space. Importantly, parts of the hidden confounder $u_t$ at time $t$ may be available to the expert but not to the imitator due to imperfect data logging or expert knowledge. We model this by segmenting the hidden confounder at time $t$ into two parts $u_t = (u_t^o, u_t^\varepsilon)$, where $u_t^o$ is observable to the expert and $u_t^\varepsilon$ is not. Intuitively, $u_t^o$ corresponds to the additional information that only the expert observes and $u_t^\varepsilon$ acts as confounding noise in the environment that affects both the state and action.[1] As a result, the transition function $\mathcal{P}(\cdot \mid s_t, a_t, (u_t^o, u_t^\varepsilon))$ at time $t$ depends on both hidden confounders, but the reward function $r(s_t, a_t, u_t^o)$ only depends on the state, action, and the observable confounder $u_t^o$ since the confounding noise only directly affects the state and actions. Finally, $\mu_0$ is the initial state distribution and $T$ is the time horizon. A causal graph illustrating these relationships is provided in Figure 1. This nuanced distinction between $u_t^o$ and $u_t^\varepsilon$ is crucial for determining the appropriate method for IL, and we begin with an example to motivate our setting and illustrate the importance of considering $u_t = (u_t^o, u_t^\varepsilon)$.

**Example 3.1.** Consider an airline ticket pricing scenario (Wright, 1928), where the goal is to learn a pricing policy by imitating actual airline pricing based on expert-set profit margins. Suppose

---

[1]In our framework, we allow the actual actions taken in the environment to be affected by the noise. Noise that only perturbs data records can be considered as a special case of our framework.

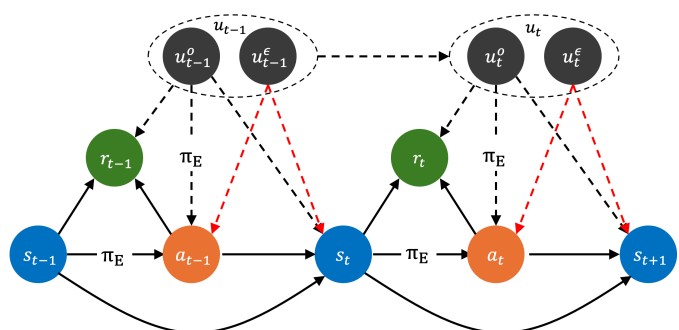

Figure 1: A causal graph of MDPs with hidden confounders $u_t = (u_t^o, u_t^\varepsilon)$. The black dashed lines represent the causal effect of the expert-observable confounder $u_t^o$, which directly affects the expert action $a_t$. It also directly affects $s_{t+1}$ and $r_t$. The red dashed lines represent the causal effect of the expert-unobservable $u_t^\varepsilon$, which acts as confounding noise and directly affects the states and actions. $u_t^\varepsilon$ does not directly affect $r_t$ (following Swamy et al. (2022b)) because the expert policy does not take $u_t^\varepsilon$ into account, and letting $u_t^\varepsilon$ directly affect $r_t$ would only add noise to the expected return.

that seasonal patterns and external events are known only to experts, but missing from the dataset. Hence, these latent variables serve as expert-observable confounders $u_t^o$. Meanwhile, actual airline prices are confounded (additively) by fluctuating operating costs, which are unknown to the experts when they set the profit margin and are not contained in the dataset. Consequently, such fluctuating operating costs act as confounding noise $u_t^\varepsilon$. We conduct experiments on a toy environment inspired by this example in Section 5, and show that IL algorithms that do *not* distinguish between $u_t^o$ and $u_t^\varepsilon$ fail to correctly imitate the expert.

**Causal Imitation Learning.** We assume that an expert is demonstrating a task following some expert policy $\pi_E$ (which we will specify in more detail later) and we observe a set of $N \geq 1$ expert demonstrations $\{d_1, d_2, ..., d_N\}$. Each demonstration is a state-action trajectory $(s_1, a_1, ..., s_T, a_T)$, where, at each time step $t$, we observe the state $s_t$ and the action $a_t$ taken in the environment. The next state is sampled from the transition function $\mathcal{P}(\ \cdot \mid s_t, a_t, (u_t^o, u_t^\varepsilon))$.

Let $h_t = (s_1, a_1, ..., s_{t-1}, a_{t-1}, s_t) \in \mathcal{H}$ denote the trajectory history at time $t$, where $\mathcal{H} \subseteq \bigcup_{i=0}^{T-1} (\mathcal{S} \times \mathcal{A})^i \times \mathcal{S}$ is the set of all possible trajectory histories. Importantly, we observe neither the reward nor the confounders $(u_t^o, u_t^\varepsilon)$ at time $t$. Given the observed trajectories, our goal is to learn a history-dependent policy $\pi_h : \mathcal{H} \to \Delta(\mathcal{A})$, where $\Delta(\mathcal{A})$ denotes the set of probability measures over $\mathcal{A}$ and the policy class $\pi_h \in \Pi$ is convex and compact. The $Q$-function of a policy $\pi_h$ is $Q_\pi(s_t, a_t, u_t^o) = \mathbb{E}_{\tau \sim \pi_h}[\sum_{t'=t}^{T} r(s_{t'}, a_{t'}, u_{t'}^o)]$ and the value of a policy is $J(\pi) = \mathbb{E}_{\tau \sim \pi_h}[\sum_{t'=1}^{T} r(s_{t'}, a_{t'}, u_{t'}^o)]$, where $\tau$ is the trajectory following $\pi_h$.

In order to learn a policy $\pi_h$ that matches the performance of $\pi_E$, we need to break the spurious correlation between states and expert actions by inferring what the expert would do if we intervened and placed them in state $s_t$ when observing $u_t^o$. Unfortunately, the causal inference literature (Shpitser & Pearl, 2008) tells us that, without further assumptions, it is generally impossible to identify $\pi_E$. To determine the minimal assumptions that allow $\pi_E$ to be identifiable, we first observe that $u_t^\varepsilon$ can be correlated for all time steps $t$, making it impossible to distinguish between the intended actions of the expert and the confounding noise. However, in practice, the confounding noise at far-apart time steps is often independent. For example, the effect of the confounding noise $u_t^\varepsilon$ at time $t$ on future states and actions often diminishes over time, which is typically the case for random environment noise such as wind. In addition, when the confounding noise $u_t^\varepsilon$ at time $t$ becomes observable at a future time $t'$, e.g., previous operating costs are observed eventually as in Example 3.1, the unobservable confounding noise at times $t$ and $t'$ becomes independent. We formalise this intuition as the notion of a *confounding noise horizon* $k$.

**Assumption 3.2** (Confounding Noise Horizon). For every $t$, the confounding noise $u_t^\varepsilon$ has a horizon of $k$ where $1 \leq k < T$. More formally, $u_t^\varepsilon \perp\!\!\!\perp u_{t-k}^\varepsilon \ \forall t > k$.

This assumption is essential for decoupling the spurious correlation between the state and action pairs. We also assume that the confounding noise is additive to the action, which is standard in causal inference (Pearl, 2000; Shao et al., 2024). Without this assumption, the causal effect becomes unidentifiable (see, e.g., Balke & Pearl (1994)) and the best we can do is to upper/lower bound it.

**Assumption 3.3** (Additive Noise). The structural equation that generates the actions in the observed trajectories is

$$a_t = \pi_E(s_t, u_t^o) + u_t^\varepsilon, \tag{2}$$

where w.l.o.g. $\mathbb{E}[u_t^\varepsilon] = 0$ as any non-zero expectation of $u_t^\varepsilon$ can be included as a constant in $\pi_E$.

Next, we show that, with the above two assumptions, it becomes possible to identify the true causal relationship between states and expert actions, and to imitate $\pi_E$.

## 4 CAUSAL IL AS A CMR PROBLEM

In this section, we demonstrate that performing causal IL in our framework is possible using trajectory histories as instruments. We show that the problem can be reformulated as a CMR problem and propose an efficient algorithm to solve it.

The typical target for IL would be the expert policy $\pi_E$ itself. However, since the expert has access to privileged information, namely $u_t^o$, which the imitator does not, the best thing an imitator can do is to learn a history-dependent policy $\pi_h$ that is the closest to the expert. A natural choice for a learning objective is the conditional expectation of $\pi_E(s_t, u_t^o)$ on the history $h_t$:

$$\pi_h(h_t) := \mathbb{E}_{\mathbb{P}(u_t^o|h_t)}[\pi_E(s_t, u_t^o)] = \mathbb{E}[\pi_E(s_t, u_t^o) \mid h_t],$$

because the conditional expectation minimises the least squares criterion (Hastie et al., 2001) and $\pi_h$ is the best predictor of $\pi_E$ given $h_t$. In $\pi_h$, the distribution $\mathbb{P}(u_t^o \mid h_t)$ captures the information about $u_t^o$ that can be inferred from trajectory histories.

**Remark 4.1.** *Learning $\pi_h$ is not trivial. Policies learnt naively using behaviour cloning, i.e., $\mathbb{E}[a_t \mid h_t]$, fail to match $\pi_E$. To see this, note that, in view of Equation (2), we have*

$$\mathbb{E}[a_t \mid h_t] = \mathbb{E}[\pi_E(s_t, u_t^o) \mid h_t] + \mathbb{E}[u_t^\varepsilon \mid h_t]$$
$$= \pi_h(h_t) + \mathbb{E}[u_t^\varepsilon \mid h_t], \tag{3}$$

*where $\mathbb{E}[u_t^\varepsilon \mid h_t] \neq 0$ due to the spurious correlation between $u_t^\varepsilon$ and the trajectory history $h_t$. As a result, $\mathbb{E}[a_t \mid h_t]$ becomes biased, which can lead to arbitrarily worse performance compared to $\pi_E$.*

**Derivation of the CMR Problem.** Leveraging the confounding horizon from Assumption 3.2, we are able to break the spurious correlation using the independence of $u_t^\varepsilon$ and $u_{t-k}^\varepsilon$. We propose to use the $k$-step history $h_{t-k} = (s_1, a_1, ..., s_{t-k})$ as an instrument for the current state $s_t$.[2] Taking the expectation conditional on $h_{t-k}$ on both sides of Equation (3) yields

$$\mathbb{E}[a_t \mid h_{t-k}] = \mathbb{E}\left[\mathbb{E}[a_t \mid h_t] \mid h_{t-k}\right] = \mathbb{E}[\pi_h(h_t) \mid h_{t-k}] + \mathbb{E}[\mathbb{E}[u_t^\varepsilon \mid h_t] \mid h_{t-k}]$$
$$= \mathbb{E}[\pi_h(h_t) \mid h_{t-k}] + \mathbb{E}[u_t^\varepsilon \mid h_{t-k}]$$
$$= \mathbb{E}[\pi_h(h_t) \mid h_{t-k}] + \mathbb{E}[u_t^\varepsilon] = \mathbb{E}[\pi_h(h_t) \mid h_{t-k}],$$

where we use the fact that $h_{t-k}$ is $\sigma(h_t)$-measurable because $h_{t-k} \subseteq h_t$, $u_t^\varepsilon \perp\!\!\!\perp u_{t-k}^\varepsilon$ and $\mathbb{E}[u_t^\varepsilon] = 0$ by Assumption 3.2. As a result, the problem of learning $\pi_h$ reduces to solving for $\pi_h$ that satisfies the following identity

$$\mathbb{E}[a_t - \pi_h(h_t) \mid h_{t-k}] = 0, \tag{4}$$

which is a CMR problem as defined in Section 2. In this case, both $a_t$ and $h_t$ are observed in the confounded expert demonstrations, and $h_{t-k}$ acts as the instrument.

To ensure that the instrument $h_{t-k}$ is valid, we formally verify that the three IV conditions from Section 2: $u_t^\varepsilon \perp\!\!\!\perp h_{t-k}$, $\mathbb{P}(h_t \mid h_{t-k})$ is not constant in $h_{t-k}$, and $h_{t-k}$ doesn't directly affect $a_t$, are satisfied by $h_{t-k}$ in Appendix B.1. However, the strength of the instrument $h_{t-k}$, representing its correlation with $h_t$, influences how well $\pi_h(h_t)$ can be identified by solving the CMR problem in Equation (4). As the confounding horizon $k$ increases, this correlation weakens, making $h_{t-k}$ a less effective instrument. We formally analyse this relationship in Proposition 4.3 and further validate it experimentally in Section 5.

---

[2]Note that this requires prior knowledge (or an upper bound) of the confounding horizon $k$. We discuss this assumption and practical ways to choose $k$, e.g., conditional independence tests, in Appendix F.

---

**Algorithm 1** Double Machine Learning for Causal Imitation Learning (DML-IL)

---

1: **input** Dataset $\mathcal{D}_E$ of expert demonstrations, confounding noise horizon $k$
2: Initialize the roll-out model $\hat{M}$ as a Gaussian mixture model
3: **repeat**
4:     Sample $(h_t, a_t)$ from data $\mathcal{D}_E$
5:     Fit the roll-out model $(h_t, a_t) \sim \hat{M}(h_{t-k})$ to maximize the log likelihood
6: **until** convergence
7: Initialize the expert model $\hat{\pi}_h$ as a neural network
8: **repeat**
9:     Sample $h_{t-k}$ from $\mathcal{D}_E$
10:     Generate $\hat{h}_t$ and $\hat{a}_t$ using the roll-out model $\hat{M}$
11:     Update $\hat{\pi}_h$ to minimise the loss $\ell := \|\hat{a}_t - \hat{\pi}_h(\hat{h}_t)\|_2$
12: **until** convergence
13: **return** A history-dependent imitator policy $\hat{\pi}_h$

---

### 4.1 Practical Algorithms for Causal IL

There are various techniques (Bennett et al., 2019a; Xu et al., 2020; Shao et al., 2024) for solving the CMR problem $\mathbb{E}[a_t|h_{t-k}] = \mathbb{E}[\pi_h(h_t)|h_{t-k}]$ in (4). Here, the *CMR error* that we aim to minimise is given by

$$\sqrt{\mathbb{E}\big[\mathbb{E}[a_t - \hat{\pi}_h(h_t)|h_{t-k}]^2\big]} = \|\mathbb{E}[a_t - \hat{\pi}_h(h_t)|h_{t-k}]\|_2.$$

In Algorithm 1, we introduce DML-IL, an algorithm adapted from the IV regression algorithm DML-IV (Shao et al., 2024), which solves our CMR problem by minimising the above CMR error.[3] The first part of the algorithm (lines 3-7) learns a roll-out model $\hat{M}$ that generates a trajectory $k$ steps ahead given $h_{t-k}$. Then, $\hat{\pi}_h$ takes the generated trajectory $\hat{h}_t$ from $\hat{M}(h_{t-k})$ as input and minimises the mean square error to the next action (lines 8-13).

Using generated trajectories is crucial for breaking the spurious correlation caused by $u_t^\varepsilon$, and the trajectory history before $h_{t-k}$ allows the imitator to infer information about $u_t^o$. In particular, the expert's future trajectory after $h_{t-k}$ is confounded with the current state and action through the unobserved noise $u_t^\varepsilon$, so it does not represent draws from the conditional distribution of future histories given $h_{t-k}$. Rolling out from $h_{t-k}$ with $\hat{M}$ removes this dependence and yields the correct conditional distribution needed for the CMR moment. We refer to Appendix F for a discussion of the theoretical convergence rate guarantees of DML-IL and the choice of the confounding noise horizon $k$ as input.

Moreover, once we set the learning objective as the conditional $\pi_h(h_t) := \mathbb{E}[\pi_E(s_t, u_t^o)|h_t]$, we can learn $\pi_h(h_t)$ for both continuous and discrete action spaces as the derivation of the CMR problem in (4) remains valid for both. However, in the algorithm and the subsequent theoretical analysis of the imitation gap, we implicitly assume that $a_t$ is continuous such that $\pi_h(h_t)$ is a valid action by the imitator. In practice, if the action space is discrete, we require a mapping that maps $\pi_h(h_t)$ to the action space, e.g., treating $\pi_h(h_t)$ as the logits output to the action space.

### 4.2 Theoretical Analysis

In this section, we derive theoretical guarantees for our algorithm, focusing on the imitation gap and its relationship to existing work. All proofs in this section are deferred to Appendix B.

On a high level, in order to bound the imitation gap of the learnt policy $\hat{\pi}_h$, i.e., $J(\pi_E) - J(\hat{\pi}_h)$, we need to control:

    ($i$) the amount of information about the hidden confounders that can be inferred from trajectory histories $h_t$;

---

[3]DML stands for double machine learning (Chernozhukov et al., 2018), which is a statistical technique to ensure a fast convergence rate for two-step regression, as is the case in Algorithm 1.

$(ii)$ the ill-posedness (or identifiability) of our CMR problem, which intuitively measures the strength of the instrument $h_{t-k}$;

$(iii)$ the disturbance of the confounding noise to the states and actions at test time.

These factors are all determined by the environment and the expert policy. To control $(i)$, we measure how much information about $u_t^o$ is captured by the trajectory history $h_t$ by analysing the Total Variation (TV) distance between the distribution of $u_t^o$ and $\mathbb{E}[u_t^o|h_t]$ along the trajectories of $\pi_E$. To control $(ii)$ and $(iii)$, we need to introduce the following two key concepts.

**Definition 4.2** (Ill-Posedness of CMRs (Dikkala et al., 2020))**.** Given the derived CMR problem in Equation (4), the *ill-posedness* $\nu(\Pi, k)$ of the policy space $\Pi$ with confounding noise horizon $k$ is

$$\nu(\Pi, k) = \sup_{\pi \in \Pi} \frac{\|\pi_E - \pi\|_2}{\|\mathbb{E}[a_t - \pi(h_t)|h_{t-k}]\|_2}.$$

The ill-posedness $\nu(\Pi, k)$ measures the strength of the instrument, where a higher $\nu(\Pi, k)$ indicates a weaker instrument. It bounds the ratio between the $L_2$ error of the imitator to the expert policy, and the learning error of the imitator following our CMR objective.

As discussed previously, intuitively, the strength of the instrument would decrease as the confounding horizon $k$ increases. This is confirmed by the following proposition.

**Proposition 4.3.** $\nu(\Pi, k)$ *is monotonically increasing as the confounded horizon $k$ increases.*

Next, we introduce the notion of c-TV stability.

**Definition 4.4** (c-Total Variation Stability (Bassily et al., 2021; Swamy et al., 2022b))**.** Let $P(X)$ be the distribution of a random variable $X : \Omega \to \mathcal{X}$. $P(X)$ is c-TV stable if for all $a_1, a_2 \in \mathcal{X}$ and $\Delta > 0$,

$$\|a_1 - a_2\| \leq \Delta \implies \delta_{TV}(a_1 + X, a_2 + X) \leq c\Delta,$$

where $\|\cdot\|$ is some norm defined on $\mathcal{X}$ and $\delta_{TV}$ is the TV distance.

A wide range of distributions are c-TV stable. For example, standard normal distributions are $\frac{1}{2}$-TV stable. We apply this notion to the distribution over $u_t^\varepsilon$ to bound the disturbance it induces in the trajectory and the expected return.

With the notion of ill-posedness and c-TV stability, we can now analyse and upper bound the imitation gap $J(\pi_E) - J(\hat{\pi}_h)$ by controlling the three previously discussed components $(i) - (iii)$.

**Theorem 4.5** (Imitation Gap Bound)**.** *Let $\hat{\pi}_h$ be the learnt policy with CMR error $\varepsilon$ and let $\nu(\Pi, k)$ be the ill-posedness of the problem. Assume that $\delta_{TV}(u_t^o, \mathbb{E}_{\pi_E}[u_t^o|h_t]) \leq \delta$ for $\delta \in \mathbb{R}^+$, $P(u_t^\varepsilon)$ is c-TV stable and $\pi_E$ is deterministic. Then, the imitation gap is upper bounded by*

$$J(\pi_E) - J(\hat{\pi}_h) \leq T^2\big(c\varepsilon\nu(\Pi, k) + 2\delta\big) = \mathcal{O}\big(T^2(\delta + \varepsilon)\big).$$

This upper bound scales at the rate of $T^2$, which aligns with the expected behaviour of imitation learning without an interactive expert (Ross & Bagnell, 2010). Next, we show that the upper bounds on the imitation gap from prior work (Swamy et al., 2022b;a) are special cases of Theorem 4.5. The proofs are deferred to Appendix B.4.

**Corollary 4.6.** *In the special case that $u_t^o = 0$, i.e., there are no expert-observable confounders, or $u_t^o = \mathbb{E}_{\pi_E}[u_t^o|h_t]$, i.e., $u_t^o$ is $\sigma(h_t)$-measurable (all information about $u_t^o$ is contained in the history), the imitation gap is upper bounded by*

$$J(\pi_E) - J(\hat{\pi}_h) \leq T^2\big(c\varepsilon\nu(\Pi, k)\big) = \mathcal{O}\big(T^2\varepsilon\big),$$

*which coincides with Theorem 5.1 of Swamy et al. (2022b).*

In the other extreme case, when there are no hidden confounders, i.e., $u_t^\varepsilon = 0$, our framework is reduced to that of Swamy et al. (2022a). However, Swamy et al. (2022a) provided an abstract bound that directly uses the supremum of key components in the imitation gap over all possible Q-functions to bound the imitation gap. We further extend and concretise the bound using the learning error $\varepsilon$ and the TV distance bound $\delta$ instead of relying on the supremum.

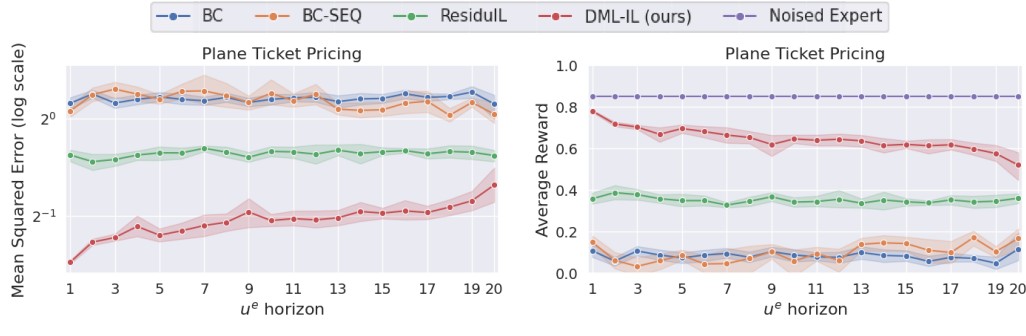

(a) MSE in log scale, lower is better.      (b) Average reward, higher is better.

Figure 2: **Plane Ticket Environment** (Example 3.1): On the left, the MSE in log scale between the learnt policy and the expert. On the right, the average reward of our approach and baselines.

**Corollary 4.7.** *In the special case that $u_t^\varepsilon = 0$, if the learnt policy has optimisation error $\varepsilon$, the imitation gap is upper bounded by*

$$J(\pi_E) - J(\hat{\pi}_h) \leq T^2 \left( \frac{2}{\sqrt{\dim(\mathcal{A})}} \varepsilon + 2\delta \right),$$

*where $\dim(\mathcal{A})$ denotes the dimension of $\mathcal{A}$. This is a concrete bound that extends the abstract bound in Theorem 5.4 of Swamy et al. (2022a).*

**Remark 4.8.** *If both $u_t^\varepsilon$ and $u_t^o$ are zero, we then recover the classic setting of IL without confounders (Ross & Bagnell, 2010), and the imitation gap bound is $T^2\varepsilon$, where $\varepsilon$ is the optimisation error of the algorithm.*

## 5 Experiments

We empirically evaluate the performance of Algorithm 1 (DML-IL) on the toy environment modelling the ticket pricing scenario with continuous state and action spaces introduced from Example 3.1 and the Mujoco environments (Todorov et al., 2012): Ant, Half Cheetah and Hopper. We compare with the following existing methods: Behavioural Cloning (BC), which naively minimises $\mathbb{E}[-\log \pi(a_t|s_t)]$; BC-SEQ (Swamy et al., 2022a), which learns a history-dependent policy to handle expert-observable hidden confounders; ResiduIL (Swamy et al., 2022b), which we here adapt to our setting by providing $h_{t-k}$ as instruments to learn a history-independent policy; and the noised expert, which is the performance of the expert in the confounded environment, and corresponds to the maximally achievable performance. In Appendix C.1, we include additional evaluations when using other IV regression algorithms, including DFIV (Xu et al., 2020) and DeepGMM (Bennett et al., 2019b), as the core CMR solver, but found inconsistent and subpar performance. In Appendix C.2, we also provide further discussion and empirical evaluations of DML-IL under misspecification of the confounding noise horizon $k$.

We train imitators with 20000 samples (40 trajectories of 500 steps each) of the expert trajectory using each algorithm and report the average reward when tested online in their respective environments. The reward is scaled such that 1 is the performance of the un-noised expert, and 0 is that of a random policy. We also report the Mean Squared Error (MSE) between the imitator's and expert's actions. The purpose of evaluating the MSE is to assess how well the imitator learnt from the expert, and importantly whether the confounding noise problem is mitigated. When the confounding noise $u_t^\varepsilon$ is explicitly handled, we should expect to observe a much higher MSE. All results are plotted with one standard deviation as a shaded area. In addition, we vary the confounding noise horizon $k$ from 1 to 20 in order to increase the difficulty of the problem with weaker instruments $h_{t-k}$.

### 5.1 Plane Ticket Pricing Environment

**Experimental Setup.** We first consider the plane ticket pricing environment described in Example 3.1. Here, the expert-unobservable confounding noise $u^\varepsilon$ corresponds to operating costs and the

expert-observable confounder $u_t^o$ models seasonal demand patterns and events. We set $u_t^o$ to continuously vary with a rate of change of approximately every 30 steps. A detailed description of this environment is provided in Appendix D.1.

**Results.** The results are presented in Figure 2. DML-IL performed best with the lowest MSE and the highest average reward that is closest to the expert, especially when the $u_t^\varepsilon$ horizon is 1. This implies that DML-IL is successful in handling both $u_t^\varepsilon$ and $u_t^o$. ResiduIL is able to reduce the confounding effect of $u_t^\varepsilon$, evident by the lower MSE compared to the two other methods that do not deal with $u_t^\varepsilon$. However, since it does not explicitly consider $u_t^o$, the imitator has no information on $u_t^o$ and the best it can do is to assume some average value (or expectation) of $u_t^o$. Therefore, while ResiduIL still achieves some reward, its considerable performance gap to DML-IL can be explained by its ignorance of $u_t^o$. Both BC and BC-SEQ fail entirely in the presence of confounding noise $u_t^\varepsilon$, with orders of magnitude higher MSE and average reward close to a random policy. From the similar performance of BC-SEQ and BC, we see that using trajectory histories to infer $u_t^o$ is not helpful when the confounding noise is not handled explicitly. This demonstrates that only partially accounting for the effect of $u_t^\varepsilon$ or $u_t^o$ is insufficient to learn a good imitator.

Moreover, as the confounding noise horizon $k$ increases (x-axis), the performance of DML-IL decreases. This supports our intuition and theoretical results that the instrument becomes weaker, and less information about $u_t^o$ can be inferred from $h_{t-k}$, as $k$ increases. When $k = 20$, we find that the performance of DML-IL is close to that of ResiduIL, which does not consider the effect of $u_t^o$, because very limited information about the current expert-observable confounder $u_t^o$ can be inferred based on the history from 20 steps ago.

## 5.2 MUJOCO ENVIRONMENTS

**Experimental Setup.** In Figure 3, we consider the Mujoco tasks. While the original environment implementations (Todorov et al., 2012) do not have hidden confounding variables, we modify the environment to introduce $u_t^\varepsilon$ and $u_t^o$. Specifically, instead of travelling as fast as possible, the goal is to control the agent to travel at a target speed that is varying throughout an episode. This target speed is $u_t^o$, which is observed by the expert but not recorded in the dataset. In addition, we add confounding noise $u_t^\varepsilon$ to $s_t$ and $a_t$ to mimic confounding noise such as wind. Additional details about the modification made to the environments are provided in Appendix D.2.

**Results.** DML-IL outperforms other methods in all three Mujoco environments as shown in Figure 3. Similarly to the plane ticket environment, ResiduIL is effective in removing the confounding noise but fails to match the average reward of DML-IL as it does not account for expert-observable confounders $u_t^o$. BC and BC-SEQ have much higher MSE and fail to learn meaningful policies. As the confounding horizon of $u_t^\varepsilon$ increases, the performance of DML-IL drops, which is expected as the instruments weaken and less information about $u_t^o$ can be inferred from the histories. This is most visible in the Ant and Half Cheetah environments.

## 6 DISCUSSION

We proposed a framework for causal imitation learning with hidden confounders that unifies several previous causal IL settings. Specifically, we considered IL from a fixed set of confounded expert demonstrations without further interactions with the confounded MDP, where the hidden confounders are partially observable to the expert. We demonstrated that causal IL under this framework can be reduced to a CMR problem when using the histories as instruments. We proposed a novel algorithm, DML-IL, to solve the CMR problem and imitate the expert, and provided upper bounds on the imitation gap of DML-IL that subsume previous results. Finally, we empirically evaluated DML-IL on multiple tasks, including Mujoco environments, and demonstrated improved imitation performance against other causal IL algorithms in the presence of expert-observable and expert-unobservable confounding.

**Limitations.** One limitation is the explicit assumptions made in Section 3, which are essential for the expert policy to be identifiable. Therefore, it is important for practitioners to validate that their specific environment and task satisfy these assumptions. We provided in the paper some examples where these assumptions are known to hold (e.g., drone and ticket sales), while we acknowledge that our method is not applicable to all scenarios, especially in the healthcare domain where non-linear

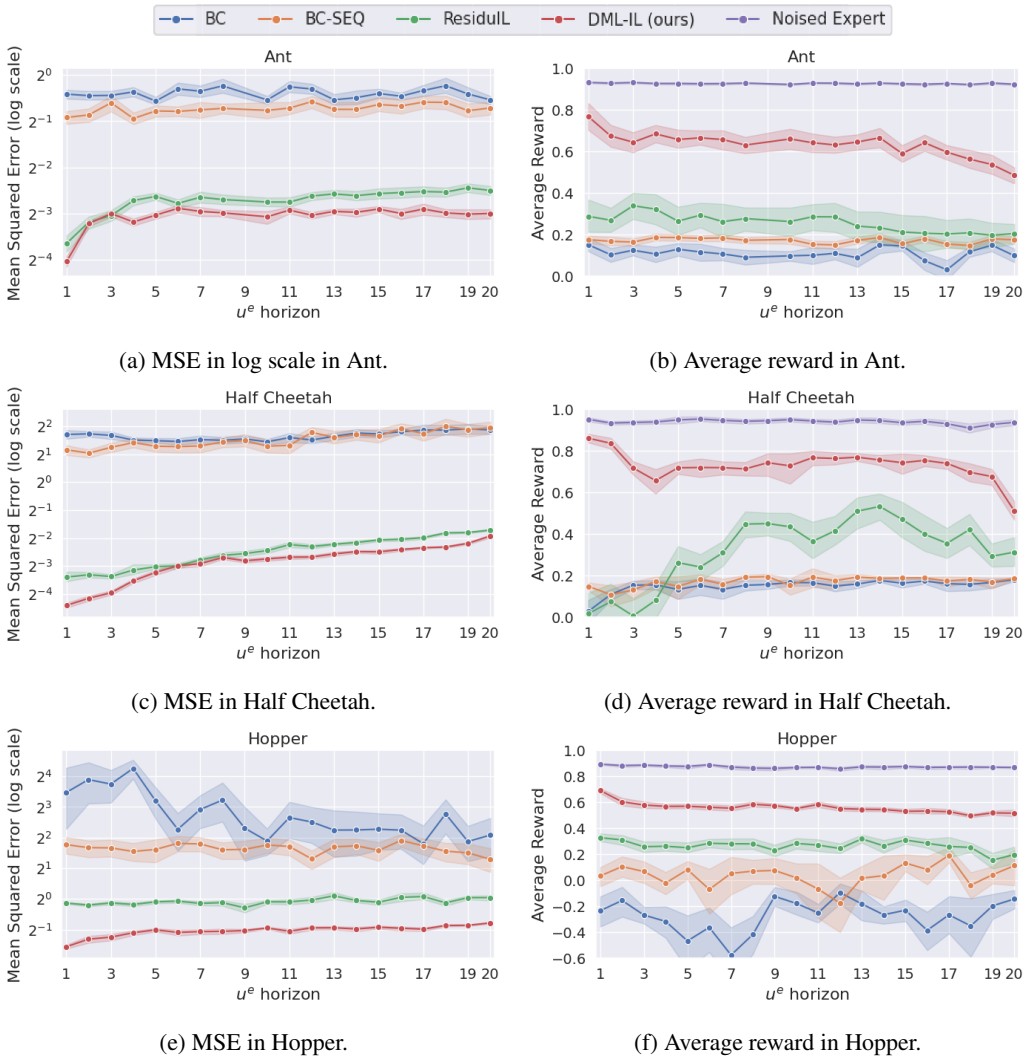

Figure 3: **MuJoCo:** On the left, the MSE in log scale between the learnt policy and the expert (lower MSE is better). On the right, the average reward in the MuJoCo environments Ant, Half Cheetah and Hopper (higher values are better). The confounding horizon increases along the x-axis.

confounding is typical. However, causal identification comes at a cost — it requires non-trivial assumptions that don't hold in all real-world applications.

In addition, we assume knowledge of the confounding noise horizon $k$ or an upper bound on it for Algorithm 1. Unfortunately, the value of $k$ generally cannot be verified empirically. However, there exist tests that can indirectly check whether a candidate IV is valid, such as conditional independence tests (Gretton et al., 2005), which we discuss in Appendix F.

**Future Works.** There are many active research fronts that consider causal identification with non-additive noise, partially observable covariates and invalid instruments. They are beyond the scope of this paper and are orthogonal to our work. It would be an interesting research direction to consider our confounded MDP framework in these problem settings.

## ACKNOWLEDGMENTS

This work was supported by the EPSRC Prosperity Partnership FAIR (grant number EP/V056883/1). DS acknowledges funding from the Turing Institute and Accenture collaboration. Part of this work was done while TKB was at The Alan Turing Institute. TKB is supported by an ETH AI Center Postdoctoral Fellowship. MK receives funding from the ERC under the European Union's Horizon 2020 research and innovation programme (FUN2MODEL, grant agreement No. 834115) and participates in Erlangen AI Hub.

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

## A    REDUCING OUR UNIFYING FRAMEWORK TO RELATED LITERATURE

In this section, we discuss how the various previous works can be obtained as special cases of our unifying framework.

### A.1    TEMPORALLY CORRELATED NOISE (SWAMY ET AL., 2022B)

The Temporally Correlated Noise (TCN) proposed in Swamy et al. (2022b) is a special case of our setting where $u^o = 0$ and only the confounding noise $u^\varepsilon$ is present. Following Equation 14-17 of Swamy et al. (2022b), their setting can be summarised as

$$
\begin{aligned}
s_t &= \mathcal{T}(s_{t-1}, a_{t-1}) \\
&= \mathcal{T}(s_{t-1}, \pi_E(s_{t-1}) + u_{t-1} + u_{t-2}) \\
a_t &= \pi_E(s_t) + u_t + u_{t-1},
\end{aligned}
$$

where $\mathcal{T}$ is the transition function and $u_t$ are the TCN. It can be seen that TCN is the confounding noise $u^\varepsilon$ since the expert policy doesn't take it into account, and it affects (or confounds) both the state and action.

It can be seen that this is a special case of our framework when $u_t^o = 0$, where $a_t = \pi_E(s_t) + \varepsilon(u_t^\varepsilon)$ from Equation (2), and more specifically when the confounding noise horizon in Theorem 3.2 is 2. In addition, the theoretical results in Swamy et al. (2022b) can be deduced from our main results as shown in Corollary 4.7.

### A.2    UNOBSERVED CONTEXTS (SWAMY ET AL., 2022A)

The setting considered by Swamy et al. (2022a) is a special case of our setting when $u^\varepsilon = 0$ and only $u^o$ are present. Following Section 3 of Swamy et al. (2022a), their setting can be summarised as

$$
\begin{aligned}
\mathcal{T} &: \mathcal{S} \times \mathcal{A} \times C \to D(S) \\
\nabla &: \mathcal{S} \times \mathcal{A} \times C \to [-1, 1] \\
a_t &= \pi_E(s_t, c)
\end{aligned}
$$

where $c \in C$ is the context, which is assumed to be fixed throughout an episode. There are no hidden confounders in this setting and the context $c$ is included in $u^o$ under our framework. Note that in our setting we also allow $u^o$ to vary throughout an episode. In addition, the theoretical results in Swamy et al. (2022a) can be deduced from our main results, as shown in Corollary 4.6.

### A.3    IMITATION LEARNING WITH LATENT CONFOUNDERS (VUORIO ET AL., 2022)

The setting considered by Vuorio et al. (2022) is also a special case of our setting when $u^\varepsilon = 0$ and only $u^o$ are present, which is very similar to Swamy et al. (2022a). In Section 2.2 of Vuorio et al. (2022), they introduced a latent variable $\theta \in \Theta$ that is fixed throughout an episode and $a_t = \pi_E(s_t, \theta)$. There are no hidden confounders in this setting and the latent variable $\theta$ is included in $u^o$ in our framework. No theoretical imitation gap bounds are provided in Vuorio et al. (2022). However, Corollary 4.6 can be directly applied to their setting and bound the imitation gap.

### A.4    CAUSAL DELUSION AND CONFUSION (ORTEGA ET AL., 2021; DE HAAN ET AL., 2019; PFROMMER ET AL., 2023; SPENCER ET AL., 2021; WEN ET AL., 2020)

The concept of causal delusion (Ortega et al., 2021) and confusion is widely studied in the literature (de Haan et al., 2019; Pfrommer et al., 2023; Spencer et al., 2021; Wen et al., 2020) from different perspectives. A classic example of causal confusion is learning to brake in an autonomous driving scenario. The states are images with a full view of the dashboard and the road conditions. The brake indicator in this scenario is the confounding variable that correlates with the action of braking in subsequent steps, which causes the imitator to learn to brake if the brake indicator light is already on. Therefore, another name for this problem is the latching problem, where the imitator latches to spurious correlations between current action and the trajectory history. In the setting of Ortega et al. (2021), this is explicitly modelled as latent variables that affect both the action and state,

causing spurious correlation between them and confusing the imitator. In other settings (de Haan et al., 2019; Pfrommer et al., 2023; Spencer et al., 2021; Wen et al., 2020), there are no explicit unobserved confounders, but the nuisance correlation between the previous states and actions can be modelled as the existence of hidden confounders $u^\varepsilon$ in our framework. Specifically, in de Haan et al. (2019), $x_{t-1}$ and $a_{t-1}$ are considered confounders that affect the state variable $x_t$, which causes a spurious correlation between previous state action pairs and $a_t$. The spurious correlation between variables is typically modelled as the existence of a hidden confounder $u^\varepsilon$ that affects both variables in causal modelling. For example, the actual hazard or event that causes the expert to brake will be the hidden confounder $u^\varepsilon$ that affects both the brake and the brake indicator.

However, despite the fact that this setting can be considered a special case of our general framework, we stress that the concrete and practical problems considered in de Haan et al. (2019); Pfrommer et al. (2023); Spencer et al. (2021); Wen et al. (2020) are different from ours, where they assumed implicitly that the hidden confounders $u^\varepsilon$ are embedded in the observations or outright observed.

## B  PROOFS OF MAIN RESULTS

In this section, we provide the proofs for the main results and corollaries in this paper.

### B.1  IV CONDITIONS FOR $h_{t-k}$

In this section, we verify that $h_{t-k}$ is a valid instrument. Firstly, we derive $u_t^\varepsilon \perp\!\!\!\perp h_{t-k}$. This follows from standard d-separation rules for causal graphs (Pearl, 2000). To establish this, we must verify that all paths from $h_{t-k} = (s_1, a_1, ..., s_{t-k})$ to $u_t^\varepsilon$ are blocked in the graph, meaning that $h_{t-k}$ is d-separated from $u_t^\varepsilon$, which implies $h_{t-k} \perp\!\!\!\perp u_t^\varepsilon$. From our causal graph in Figure 1, we see that any paths from $h_{t-k}$ to $u_t^\varepsilon$ must pass through a collider structure, specially through either $s_t \to a_t \leftarrow u_t^\varepsilon$ or $a_t \to s_{t+1} \leftarrow u_t^\varepsilon$. Furthermore, potential paths through hidden confounders are ruled out because there are no direct causal paths between $u_{t-k}^\varepsilon$ and $u_t^\varepsilon$, as required by Assumption 3.2. Thus, all paths from $h_{t-k}$ to $u_t^\varepsilon$ are blocked by d-separation, and we can conclude that $h_{t-k} \perp\!\!\!\perp u_t^\varepsilon$. Secondly, $\mathbb{P}(h_t \mid h_{t-k})$ is not constant in $h_{t-k}$ because we can assume that the environment is non trivial and the past state have an impact on future states. Finally, $h_{t-k}$ doesn't directly affect $a_t$, specifically $h_{t-k} \perp\!\!\!\perp a_t \mid (s_t, u_t^\varepsilon, u_t^o)$, by the Markov property — the next action $a_t$ and the trajectory history are conditionally independent given the current state $s_t$.

### B.2  PROOF OF PROPOSITIONS

**Proposition 4.3**: The ill-posedness $\nu(\Pi, k)$ is monotonically increasing as the confounded horizon $k$ increases.

*Proof.* From definition, we have that

$$\nu(\Pi, k) = \sup_{\pi \in \Pi} \frac{\|\pi_E - \pi\|_2}{\|\mathbb{E}[a_t - \pi(h_t)|h_{t-k}]\|_2}.$$

We would like to show for each $\pi \in \Pi$, $\frac{\|\pi_E - \pi\|_2}{\|\mathbb{E}[a_t - \pi(h_t)|h_{t-k}]\|_2}$ is increasing as $k$ increases, which would imply that $\nu(\Pi, k)$ is increasing. For each $\pi \in \Pi$, we see that the numerator is constant w.r.t the horizon $k$. Therefore, it is enough to check that for each $\pi \in \Pi$, the denominator $\|\mathbb{E}[a_t - \pi(h_t)|h_{t-k}]\|_2$ decreases as $k$ increases. For any two integer horizon $k_1 > k_2$,

$$\mathbb{E}[a_t - \pi(h_t)|h_{t-k_1}]^2 = \mathbb{E}[\mathbb{E}[a_t - \pi(h_t)|h_{t-k_2}]|h_{t-k_1}]^2 \tag{5}$$

$$\leq \mathbb{E}[\mathbb{E}[a_t - \pi(h_t)|h_{t-k_2}]^2|h_{t-k_1}] \tag{6}$$

$$= \mathbb{E}[a_t - \pi(h_t)|h_{t-k_2}]^2 \tag{7}$$

by the tower property of conditional expectation as $\sigma(h_{t-k_1}) \subseteq \sigma(h_{t-k_2})$, Jensen's inequality for conditional expectations, and the fact that $\mathbb{E}[a_t - \pi(h_t)|h_{t-k_2}]^2$ is $h_{t-k_1}$ measurable, respectively for each line. Therefore, we have that $\mathbb{E}[a_t - \pi(h_t)|h_{t-k}]$ is decreasing, which implies $\|\mathbb{E}[a_t - \pi(h_t)|h_{t-k}]\|_2$ is decreasing and $\nu(\Pi, k)$ is increasing as $k$ increases, which completes the proof. $\square$

## B.3 Main results for guarantees on the imitation gap

**Theorem 4.5**: Let $\hat{\pi}_h$ be the learnt policy with CMR error $\varepsilon$ and let $\nu(\Pi, k)$ be the ill-posedness of the problem. Assume that $\delta_{TV}(u_t^o, \mathbb{E}_{\pi_E}[u_t^o|h_t]) \leq \delta$ for $\delta \in \mathbb{R}^+$, $P(u_t^\varepsilon)$ is c-TV stable and $\pi_E$ is deterministic. Then, the imitation gap is upper bounded by

$$J(\pi_E) - J(\hat{\pi}_h) \leq T^2(c\varepsilon\nu(\Pi, k) + 2\delta) = \mathcal{O}(T^2(\delta + \varepsilon)).$$

*Proof of Theorem 4.5.* Recall that $J(\pi)$ is the expected reward following $\pi$, and we would like to bound the performance gap $J(\pi_E) - J(\hat{\pi}_h)$ between the expert policy $\pi_E$ and the learned history-dependent policy $\hat{\pi}_h$. Let $Q_{\hat{\pi}_h}(s_t, a_t, u_t^o)$ be the Q-function of $\hat{\pi}_h$. Using the Performance Difference Lemma (Kakade & Langford, 2002), we have that for any Q-function $\tilde{Q}(h_t, a_t)$ that takes in the trajectory history $h_t$ and action $a_t$,

$$J(\pi_E) - J(\hat{\pi}_h) = \mathbb{E}_{\tau \sim \pi_E}[\sum_{t=1}^T Q_{\hat{\pi}_h}(s_t, a_t, u_t^o) - \mathbb{E}_{a \sim \hat{\pi}_h}[Q_{\hat{\pi}_h}(s_t, a, u_t^o)]]$$

$$= \sum_{t=1}^T \mathbb{E}_{\tau \sim \pi_E}[Q_{\hat{\pi}_h}(s_t, a_t, u_t^o) - \tilde{Q}(h_t, a_t) + \tilde{Q}(h_t, a_t) - \mathbb{E}_{a \sim \hat{\pi}_h}[Q_{\hat{\pi}_h} - \tilde{Q} + \tilde{Q}]]$$

$$= \sum_{t=1}^T \mathbb{E}_{\tau \sim \pi_E}[\tilde{Q} - \mathbb{E}_{a \sim \hat{\pi}_h}[\tilde{Q}]] + \sum_{t=1}^T \mathbb{E}_{\tau \sim \pi_E}[Q_{\hat{\pi}_h} - \tilde{Q} - \mathbb{E}_{a \sim \hat{\pi}_h}[Q_{\hat{\pi}_h} - \tilde{Q}]] \quad (8)$$

We first bound the second part of Equation (8). Denote by $\delta_{TV}$ the total variation distance. For two distributions $P, Q$, recall the property of total variation distance for bounding the difference in expectations:

$$|\mathbb{E}_P[f(x)] - \mathbb{E}_Q[f(x)]| \leq \|f\|_\infty \delta_{TV}(P, Q).$$

In order to bound the second part of Equation (8), for any $Q$ function, consider inferred $\tilde{Q}$ using the conditional expectation of $u^o$ on the history $h$,

$$\tilde{Q}(h_t, a_t) := Q(s_t, a_t, \mathbb{E}_{\tau \sim \pi_E}[u_t^o|h_t]),$$

where note that $s_t \in h_t$. We have that, when the transition trajectory $(s_t, u_t^o, u_t^\varepsilon, r_t) \sim \pi_E$ follows the expert policy, for any action $\dot{a} \sim \pi$ following some policy $\pi$ (in our case, it can be $\pi_E$ or $\hat{\pi}_h$),

$$|\mathbb{E}_{\tau \sim \pi_E, \dot{a} \sim \pi}[Q(s_t, \dot{a}, u_t) - \tilde{Q}(h_t, \dot{a})]| = |\mathbb{E}_{\tau \sim \pi_E, \dot{a} \sim \pi}[Q(s_t, \dot{a}, u_t^o) - Q(s_t, \dot{a}, \mathbb{E}_{\tau \sim \pi_E}[u_t^o|h_t])]|$$

$$= |\mathbb{E}_{u_t^o \sim \pi_E}[\mathbb{E}_{\pi_E, \pi}[Q(s_t, \dot{a}, u_t^o)|u_t^o] - \mathbb{E}_{u_t^o|h_t \sim \pi_E}[\mathbb{E}_{\pi_E, \pi}[Q(s_t, \dot{a}, u_t^o)|u_t^o]| \quad (9)$$

$$\leq \|\mathbb{E}_{\pi_E, \pi}[Q(s_t, \dot{a}, u_t^o)|u_t^o]\|_\infty \delta_{TV}(u_t^o, \mathbb{E}_{\pi_E}[u_t^o|h_t]) \quad (10)$$

$$\leq T \cdot \delta_{TV}(u_t^o, \mathbb{E}_{\pi_E}[u_t^o|h_t]) \quad (11)$$

$$\leq T\delta \quad (12)$$

where Equation (9) uses the tower property of expectations, Equation (10) uses the total variation distance bound for bounded functions, Equation (11) uses the fact that the $Q$ function is bounded by $T$ and Equation (12) uses the condition that $\delta_{TV}(u_t^o, \mathbb{E}_{\pi_E}[u_t^o|h_t]) \leq \delta$ in the theorem statement. Since Equation (8) holds for any choice of $\tilde{Q}$, we choose $\tilde{Q}_{\hat{\pi}_h}(h_t, a_t) := Q_{\hat{\pi}_h}(s_t, a_t, \mathbb{E}_{\tau \sim \pi_E}[u_t^o|h_t])$ such that we can apply Equation (12) twice to bound the second part of Equation (8):

$$\mathbb{E}_{\tau \sim \pi_E}[Q_{\hat{\pi}_h} - \tilde{Q}_{\hat{\pi}_h} - \mathbb{E}_{a \sim \hat{\pi}_h}[Q_{\hat{\pi}_h} - \tilde{Q}_{\hat{\pi}_h}]] \leq \mathbb{E}_{\tau \sim \pi_E}[Q_{\hat{\pi}_h} - \tilde{Q}_{\hat{\pi}_h} + |\mathbb{E}_{a \sim \hat{\pi}_h}[Q_{\hat{\pi}_h} - \tilde{Q}_{\hat{\pi}_h}]|]$$

$$= \mathbb{E}_{\tau \sim \pi_E}[Q_{\hat{\pi}_h} - \tilde{Q}_{\hat{\pi}_h}] + |\mathbb{E}_{s_t, u_t \sim \pi_E, a \sim \hat{\pi}_h}[Q_{\hat{\pi}_h} - \tilde{Q}_{\hat{\pi}_h}]|$$

$$\leq |\mathbb{E}_{\tau \sim \pi_E}[Q_{\hat{\pi}_h} - \tilde{Q}_{\hat{\pi}_h}]| + T\delta \quad (13)$$

$$\leq 2T\delta$$

where Equation (13) holds by applying Equation (12) because the expectation of the trajectories (and their transitions) are over $\pi_E$, and the actions which are used only as arguments into the $Q$ function are sampled from $\hat{\pi}_h$.

Next, we bound the first part of Equation (8). Recall that the ill-posedness of the problem for a policy class $\Pi$ is

$$\nu(\Pi, k) = \sup_{\pi \in \Pi} \frac{\|\pi_E - \pi\|_2}{\|\mathbb{E}[a_t - \pi(h_t)|h_{t-k}]\|_2}$$

where $\|\pi_E - \pi\|_2$ is the RMSE and $\|\mathbb{E}[a_t - \pi(s_t)|s_{t-k}]\|_2$ is the CMR error from our algorithm. Since the learned policy $\hat{\pi}_h$ has a CMR error of $\varepsilon$, we have that

$$\|\pi_E - \hat{\pi}_h\|_2 \leq \nu(\Pi, k)\|\mathbb{E}[a_t - \hat{\pi}_h(h_t)|h_{t-k}]\|_2 \leq \nu(\Pi, k)\varepsilon$$

Next, recall that c-total variation stability of a distribution $P(u^\varepsilon)$ where $u^\varepsilon \in A$ for some space $A$ implies for two elements $a_1, a_2 \in A$,

$$\|a_1 - a_2\|_2 \leq \Delta \implies \delta_{TV}(a_1 + u^\varepsilon, a_2 + u^\varepsilon) \leq c\Delta.$$

Since $P(u_t^\varepsilon)$ is c-TV stable w.r.t the action space $A$, we have that for all history trajectories $h_t \in H$ (note that $s_t \in h_t$)

$$\delta_{TV}(\pi_E(s_t) + u_t^\varepsilon, \hat{\pi}_h(h_t) + u_t^\varepsilon) \leq c\|\pi_E(s_t) - \hat{\pi}_h(h_t)\|_2.$$

Then, we have that by Jensen's inequality,

$$\mathbb{E}_{h_t \sim \pi_E}[\delta_{TV}(\pi_E(s_t) + u_t^\varepsilon, \hat{\pi}_h(h_t) + u_t^\varepsilon)]^2 \leq \mathbb{E}_{h_t \sim \pi_E}[\delta_{TV}(\pi_E(s_t) + u_t^\varepsilon, \hat{\pi}_h(h_t) + u_t^\varepsilon)^2]$$

$$\implies \mathbb{E}_{h_t \sim \pi_E}[\delta_{TV}(\pi_E(s_t) + u_t^\varepsilon, \hat{\pi}_h(h_t) + u_t^\varepsilon)] \leq \sqrt{\mathbb{E}_{h_t \sim \pi_E}[\delta_{TV}(\pi_E(s_t) + u_t^\varepsilon, \hat{\pi}_h(h_t) + u_t^\varepsilon)^2]}$$

$$\leq \sqrt{c^2 \mathbb{E}_{h_t \sim \pi_E}[\|\pi_E(s_t) - \hat{\pi}_h(h_t)\|_2^2]}$$

$$= c\|\pi_E - \hat{\pi}_h\|_2 \leq c\varepsilon\nu(\Pi, k)$$

Therefore, by applying the total variation distance bound for expectations of $\tilde{Q}_{\hat{\pi}_h}$ over different distributions of action $a_t$, we have that

$$\mathbb{E}_{\tau \sim \pi_E}[\tilde{Q}_{\hat{\pi}_h} - \mathbb{E}_{a \sim \hat{\pi}_h}[\tilde{Q}_{\hat{\pi}_h}]] = \mathbb{E}_{\tau \sim \pi_E}[\tilde{Q}_{\hat{\pi}_h}(h_t, a_t) - \mathbb{E}[\tilde{Q}_{\hat{\pi}_h}(h_t, \hat{\pi}_h(h_t))]] \tag{14}$$

$$= \mathbb{E}_{h_t \sim \pi_E}[\mathbb{E}[\tilde{Q}_{\hat{\pi}_h}(h_t, \pi_E(s_t) + u_t^\varepsilon)] - \mathbb{E}[\tilde{Q}_{\hat{\pi}_h}(h_t, \hat{\pi}_h(h_t) + u_t^\varepsilon)]] \tag{15}$$

$$\leq \|\tilde{Q}_{\hat{\pi}_h}\|_\infty \mathbb{E}_{h_t \sim \pi_E}[\delta_{TV}(F(\pi_E(s_t) + u_t^\varepsilon), F(\hat{\pi}_h(h_t) + u_t^\varepsilon))] \tag{16}$$

$$\leq Tc\varepsilon\nu(\Pi, k) \tag{17}$$

Combining all of above, we see that from Equation (8), by selecting $\tilde{Q}_{\hat{\pi}_h}(h_t, a_t) := Q_{\hat{\pi}_h}(s_t, a_t, \mathbb{E}_{\tau \sim \pi_E}[u_t^o|h_t])$, the imitation gap can be bounded by

$$J(\pi_E) - J(\hat{\pi}_h) = \sum_{t=1}^{T} \mathbb{E}_{\tau \sim \pi_E}[\tilde{Q}_{\hat{\pi}_h} - \mathbb{E}_{a \sim \hat{\pi}_h}[\tilde{Q}_{\hat{\pi}_h}]] + \sum_{t=1}^{T} \mathbb{E}_{\tau \sim \pi_E}[Q_{\hat{\pi}_h} - \tilde{Q}_{\hat{\pi}_h} - \mathbb{E}_{a \sim \hat{\pi}_h}[Q_{\hat{\pi}_h} - \tilde{Q}_{\hat{\pi}_h}]] \tag{18}$$

$$\leq \sum_{t=1}^{T} Tc\varepsilon\nu(\Pi, k) + \sum_{t=1}^{T} 2T\delta \tag{19}$$

$$\leq T \cdot (Tc\varepsilon\nu(\Pi, k) + 2T\delta) \tag{20}$$

$$= T^2(c\varepsilon\nu(\Pi, k) + 2\delta) = \mathcal{O}(T^2(\varepsilon + \delta)), \tag{21}$$

which concludes the proof. $\qquad\square$

### B.4 PROOFS OF COROLLARIES

**Corollary 4.6:** In the special case that $u_t^o = 0$, meaning that there is no confounder observable to the expert, or $u_t^o = \mathbb{E}_{\pi_E}[u_t^o|h_t]$, meaning that $u_t^o$ is $\sigma(h_t)$ measurable (all information regarding $u_t^o$ is represented in the history), the imitation gap bound is $T^2(c\varepsilon\nu(\Pi, k))$, which coincides with Theorem 5.1 of Swamy et al. (2022b).

*Proof.* If $u_t^o = 0$, then we have $u_t^o = \mathbb{E}_{\pi_E}[u_t^o|h_t]$ since $u_t^o$ is a constant. If $u_t^o = \mathbb{E}_{\pi_E}[u_t^o|h_t]$, we have that

$$\delta_{TV}(u_t^o, \mathbb{E}_{\pi_E}[u_t^o|h_t]) = \delta_{TV}(u_t^o, u_t^o) \leq 0$$

By plugging $\delta = 0$ into Theorem 4.5, we have that $J(\pi_E) - J(\hat{\pi}_h) \leq T^2(c\varepsilon\nu(\Pi, k))$, which is the same as the imitation gap derived in Swamy et al. (2022b) and completes the proof. $\square$

**Corollary 4.7:** In the special case that $u_t^\varepsilon = 0$, if the learned policy via supervised BC has error $\varepsilon$, then the imitation gap bound is $T^2(\frac{2}{\sqrt{\dim(A)}}\varepsilon + 2\delta)$, which is a concrete bound that extends the abstract bound in Theorem 5.4 of Swamy et al. (2022a).

*Proof.* In Theorem 5.4 of Swamy et al. (2022a), for the offline case, which is the setting we are considering (as opposed to the online settings), they defined the following quantities for bounding the imitation gap in a very general fashion,

$$\varepsilon_{\text{off}} := \sup_{\tilde{Q}} \mathbb{E}_{\tau \sim \pi_E}[\tilde{Q} - \mathbb{E}_{a \sim \hat{\pi}_h}[\tilde{Q}]]$$

$$\delta_{\text{off}} := \sup_{Q \times \tilde{Q}} \mathbb{E}_{\tau \sim \pi_E}[Q_{\hat{\pi}_h} - \tilde{Q} - \mathbb{E}_{a \sim \hat{\pi}_h}[Q_{\hat{\pi}_h} - \tilde{Q}]].$$

The imitation gap by Theorem 5.4 in Swamy et al. (2022a) under the assumption that $u_t^\varepsilon = 0$ is $T^2(\varepsilon_{\text{off}} + \delta_{\text{off}})$, which can also be deduced from Equation (8) by naively applying the above supremum. To obtain a concrete bound, we can provide a tighter bound for $\mathbb{E}_{\tau \sim \pi_E}[\tilde{Q}_{\hat{\pi}_h} - \mathbb{E}_{a \sim \hat{\pi}_h}[\tilde{Q}_{\hat{\pi}_h}]]$, which is the first part of Equation (8), given that $u_t^\varepsilon = 0$.

For two elements $a_1, a_2 \in A$, we have that by Cauchy–Schwarz,

$$\delta_{TV}(a_1 + 0, a_2 + 0) = \frac{1}{2}\|a1 - a2\|_1 \leq \frac{\sqrt{\dim(A)}}{2}\|a1 - a2\|_2.$$

Then, we have that

$$\|a_1 - a_2\|_2 \leq \Delta \implies \delta_{TV}(a_1, a_2) \leq \frac{2}{\sqrt{\dim(A)}}\Delta$$

so that by Theorem 4.5,

$$\mathbb{E}_{\tau \sim \pi_E}[\tilde{Q}_{\hat{\pi}_h} - \mathbb{E}_{a \sim \hat{\pi}_h}[\tilde{Q}_{\hat{\pi}_h}]] = \mathbb{E}_{\tau \sim \pi_E}[\tilde{Q}_{\hat{\pi}_h}(h_t, a_t) - \mathbb{E}[\tilde{Q}_{\hat{\pi}_h}(h_t, \hat{\pi}_h(h_t))]] \tag{22}$$

$$= \mathbb{E}_{h_t \sim \pi_E}[\mathbb{E}[\tilde{Q}_{\hat{\pi}_h}(h_t, \pi_E(s_t))] - \mathbb{E}[\tilde{Q}_{\hat{\pi}_h}(h_t, \hat{\pi}_h(h_t))]] \tag{23}$$

$$\leq \|\tilde{Q}_{\hat{\pi}_h}\|_\infty \frac{2}{\sqrt{\dim(A)}}\|\pi_E - \pi\|_2 \tag{24}$$

$$\leq T\frac{2}{\sqrt{\dim(A)}}\varepsilon, \tag{25}$$

since when $u_t^\varepsilon = 0$ the learning error via supervised learning is $\varepsilon := \|\pi_E - \pi\|_2$. Therefore, the final imitation bound following Theorem 4.5 is

$$J(\pi_E) - J(\hat{\pi}_h) = \sum_{t=1}^{T} \mathbb{E}_{\tau \sim \pi_E}[\tilde{Q}_{\hat{\pi}_h} - \mathbb{E}_{a \sim \hat{\pi}_h}[\tilde{Q}_{\hat{\pi}_h}]] + \sum_{t=1}^{T} \mathbb{E}_{\tau \sim \pi_E}[Q_{\hat{\pi}_h} - \tilde{Q}_{\hat{\pi}_h} - \mathbb{E}_{a \sim \hat{\pi}_h}[Q_{\hat{\pi}_h} - \tilde{Q}_{\hat{\pi}_h}]] \tag{26}$$

$$\leq \sum_{t=1}^{T} T\frac{2}{\sqrt{\dim(A)}}\varepsilon + \sum_{t=1}^{T} 2T\delta \tag{27}$$

$$= T^2(\frac{2}{\sqrt{\dim(A)}}\varepsilon + 2\delta). \tag{28}$$

This bound is a concrete bound, obtained through detailed analysis of the problem at hand, that coincides with the abstract bound $T^2(\varepsilon_{\text{off}} + \delta_{\text{off}})$ provided in Theorem 5.4 of Swamy et al. (2022b). Note that this bound is independent of the ill-posedness $\nu(\Pi, k)$ and the c-TV stability of $u_t^{\bar{\varepsilon}}$, which are present in the bound of Theorem 4.5, because of the lack of hidden confounders $u_t^\varepsilon$. $\square$

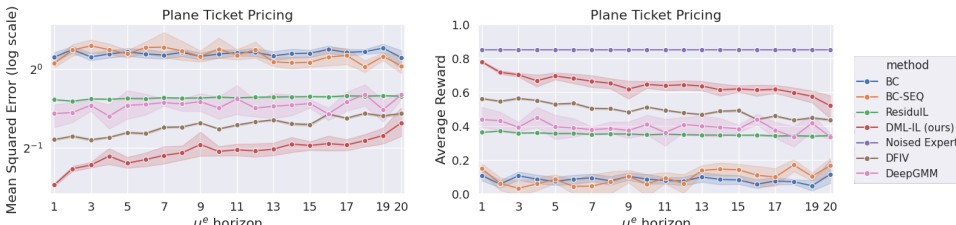

Figure 4: Additional results for the MSE between learnt policy and expert, and the average reward, in the plane ticket environment (Example 3.1), with DFIV and DeepGMM as the CMR solver.

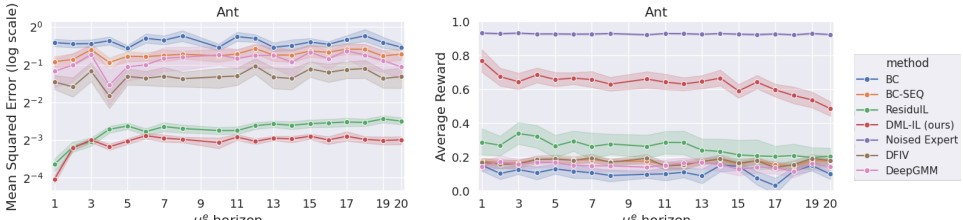

Figure 5: Additional results for the MSE between learnt policy and expert, and the average reward, Ant Mujoco environment, with DFIV and DeepGMM as the CMR solver.

## C    ADDITIONAL EXPERIMENTS

### C.1    ADOPTING OTHER IV REGRESSION ALGORITHMS

In this paper, we have transformed causal IL with hidden confounders into a CMR problem as defined in Equation (4). Therefore, in principle, many IV regression algorithms can be adopted to solve our CMR problem. We also experimented with other IV regression algorithms that have been previously shown to be practical (Shao et al., 2024) for different tasks and high-dimensional input. Specifically, we experimented with DFIV (Xu et al., 2020), which is an iterative algorithm that integrates the training of two models that depend on each other, and DeepGMM (Bennett et al., 2019b), which solves a minimax game by optimising two models adversarially. Note that DeepIV (Hartford et al., 2017) can be considered a special case of DML-IV (Shao et al., 2024), so we did not evaluate it.

The additional results for using DFIV and DeepGMM as the CMR solver are provided in Figure 4 and Figure 5. It can be seen from Figure 4 that only DFIV achieves good performance in the airline ticket pricing environment, surpassing the performance of ResiduIL. For the Ant Mujoco task in Figure 5, both DFIV and DeepGMM fail to learn good policies, with only slightly lower MSE than BC and BC-SEQ. We think this is mainly due to the high-dimensional state and action spaces and the inherent instability in the DFIV and DeepGMM algorithms. For DFIV, the interleaving of training of two models causes highly non-stationary training targets for both models, and, for DeepGMM, the adversarial training procedure of two models is similar to that of generative adversarial networks (GANs), which are known to be unstable and difficult to train. In addition, when the CMR problem is weakly identifiable, as in the case of a weak instrument, the algorithms may converge to local minima that are far away from the true solution in the face of instabilities in the algorithm.

We conclude that solving our CMR problem can be sensitive to the choice of solver as well as to the choice of hyperparameters. In addition, some IV regression algorithms do not work well with high-dimension inputs. Our IV algorithm of choice, DML-IV, provides a robust base for the DML-IL algorithm that demonstrated good performance across all tasks and environments. This demonstrates the benefit of using double machine learning, which can debias two-stage estimators and provide good empirical and theoretical convergence.

## C.2 Performance under misspecification of $k$

When past unobservable confounders $u_{t-k}^{\varepsilon}$ are weakly correlated with the current $u_t^{\varepsilon}$, the unconfounded instrument condition for a valid IV is mildly violated. Empirically, when the violation is mild, it typically induces small bias. This is especially true if the correlation between the IV and hidden confounder is weak relative to IV strength (Hahn & Hausman, 2005), i.e., the correlation between $h_{t-k}$ and the current state $s_t$. It is also often observed that there is a threshold effect (Kuang et al., 2020), where once the violation rises above a certain threshold, IV regression begins to induce large bias.

However, to the best of our knowledge, there is no theoretical framework that can analyse IV regression bias with respect to IV violation with guarantees. In fact, in a theoretical worst-case, a weak correlation between the IV and the hidden confounder could potentially cause the causal effect to be unidentifiable, rendering causal inference tools ineffective.

That being said, there also exist methods that can combine weak or mildly invalid IVs to synthesise valid IVs (Kuang et al., 2020; Hartford et al., 2020; Yuan et al., 2022) and it would be possible to combine the trajectory history $h_{t-k}$, which may contain invalid IVs, into a valid IV.

To empirically evaluate this, we conduct additional experiments where the true confounding horizon is 10, but DML-IL is given the misspecified $k = 1$ to 9. With $k = 10$ as the baseline without misspecification, performance (avg reward) in Half Cheetah stays within $5\%$ of the baseline down to $k = 6$, and remains acceptable down to $k = 8$ in the plane ticket task, after which DML-IL starts to induce larger bias. We report the average reward together with its standard deviation (in parentheses).

| Misspecified k | Half Cheetah | Plane Ticket |
|:---:|:---:|:---:|
| k=10 (no misspecification) | 0.7183 (0.1789) | 0.6181 (0.0356) |
| k=9 | 0.7108 (0.1193) | 0.5973 (0.0242) |
| k=8 | 0.7209 (0.1717) | 0.5546 (0.0325) |
| k=7 | 0.6675 (0.1595) | 0.4801 (0.0614) |
| k=6 | 0.6903 (0.1393) | 0.3944 (0.0682) |
| k=5 | 0.3471 (0.1989) | 0.3241 (0.0773) |
| k=4 | 0.3243 (0.2329) | 0.1561 (0.0961) |
| k=3 | 0.2749 (0.1643) | 0.1076 (0.1310) |
| k=2 | 0.1082 (0.2155) | 0.0801 (0.1469) |
| k=1 | 0.0896 (0.3080) | 0.0656 (0.1227) |

Table 1: Performance across misspecified $k$ values for Half Cheetah and Plane Ticket.

## D Environments and Tasks

### D.1 Dynamic Aeroplane Ticket Pricing

Here, we provide details regarding the dynamic aeroplane ticket pricing environment introduced in Example 3.1. The environment and the expert policy are defined as follows:

$$\mathcal{S} := \mathbb{R} \tag{29}$$
$$\mathcal{A} := [-1, 1] \tag{30}$$
$$s_t = sign(s) \cdot u_t^o - u_t^{\varepsilon} \tag{31}$$
$$\pi_E = clip(-s/u_t^o, -1, 1) \tag{32}$$
$$a_t = \pi_E + 10 \cdot u_t^{\varepsilon} \tag{33}$$
$$u_t^o = mean(p_t \sim \text{Unif}[-1, 1], p_{t-1}, ....p_{t-M}) \tag{34}$$
$$u_t^{\varepsilon} = mean(q_t \sim \text{Normal}(0, 0.1 \cdot \sqrt{k}), q_{t-1}, ..., q_{t-k+1}) \tag{35}$$

where $M$ is the influence horizon of the expert-observable $u^o$, which we set to 30. The states $s_t$ are the profits at each time step, and the actions $a_t$ are the final ticket price. $u_t^o$ represent the seasonal patterns, where the expert $\pi_E$ will try to adjust the price accordingly. $u_t^{\varepsilon}$ represent the operating

costs, which are additive both to the profit and price. Both $u_t^o$ and $u_t^\varepsilon$ are the mean over a set of i.i.d samples, $q_t$ and $p_t$, and vary across the time steps by updating the elements in the set at each time step. This construction allows $u_t^\varepsilon$ and $u_{t-k}^\varepsilon$ to be independent since all set elements $q_t$ will be re-sampled from time step $t - k$ to $t$. We multiply the standard deviation of $q_t$ by $\sqrt{k}$ to make sure $u_t^\varepsilon$, which is the average over $k$ i.i.d. variables, has the same standard deviation for all choices of $k$.

## D.2 MUJOCO ENVIRONMENTS

We evaluate DML-IL on three Mujoco environments: Ant, Half Cheetah, and Hopper. The original tasks do not contain hidden variables, so we modify the environment to introduce $u^\varepsilon$ and $u^o$. We use the default transition, state, and action space defined in the Mujoco environment. However, we changed the task objectives by altering the reward function and added confounding noise to both the state and action. Specifically, instead of controlling the ant, half cheetah, and hopper, respectively, to travel as fast as possible, the goal is to control the agent to travel at a target speed that is varying throughout an episode. This target speed is $u^o$, which is observed by the expert but not recorded in the dataset. In addition, we add confounding noise $u_t^\varepsilon$ to $s_t$ and $a_t$ to mimic the environmental noise such as wind noise. In all cases, the target speed $u_t^o$, confounding noise $u_t^\varepsilon$, and the action $a_t$ are generated as follows:

$$a_t = \pi_E + 20 \cdot u_t^\varepsilon \tag{36}$$

$$u_t^o = mean(p_t \sim \text{Unif}[-2, 4], p_{t-1}, ....p_{t-M}) \tag{37}$$

$$u_t^\varepsilon = mean(q_t \sim \text{Normal}(0, 0.01 \cdot \sqrt{k}), q_{t-1}, ..., q_{t-k+1}) \tag{38}$$

where $M = 30$, the state transitions follow the default Mujoco environment and the expert policy $\pi_E$ is learned online in the environment. $u_t^o$ and $u_t^\varepsilon$ follow the aeroplane ticket pricing environment to be the average over a queue of i.i.d. random variables. The reward is defined to be the $1_{healthy} - $ (current velocity $- u_t^o)^2 - $ control loss, where $1_{healthy}$ gives reward 1 as long as the agent is in a healthy state as defined in the Mujoco documentation. The second penalty term penalises deviation between the current agent's velocity and the target velocity $u_t^o$. The control loss term is also as defined in default Mujoco, which is $0.1 * \sum(a_t^2)$ at each step to regularize the size of actions.

### D.2.1 ANT

In the Ant environment, we follow the gym implementation [4] with an 8-dimensional action space and a 28-dimensional observable state space, where the agent's position is also included in the state space. Since the target speed $u_t^o$ is not recorded in the trajectory dataset, we scale the current position of the agent with respect to the target speed, $pos_t' = pos_{t-1} + \frac{pos_t - pos_{t-1}}{u_t^o}$, and use the new agent position $pos_t'$ in the observed states. This allows the imitator to infer information regarding $u_t^o$ from trajectory history, namely from the rate of change in the past positions.

### D.2.2 HALF CHEETAH

In the Half Cheetah environment, we follow the gym implementation [5] with a 6-dimensional action space and an 18-dimensional observable state space, where the agent's position is also included in the state space. Similarly to the Ant environment, we scale the current position of the agent to $pos_t' = pos_{t-1} + \frac{pos_t - pos_{t-1}}{u_t^o}$ such that the imitator can infer information regarding $u_t^o$ from trajectory history.

### D.2.3 HOPPER

In the Hopper environment, we follow the gym implementation [6] with a 3-dimensional action space and a 12-dimensional observable state space, where the agent's position is also included in the state space. Similarly to the Ant environment, we scale the current position of the agent to $pos_t' = $

---

[4]Ant environment: https://www.gymlibrary.dev/environments/mujoco/ant/

[5]Half Cheetah environment: https://www.gymlibrary.dev/environments/mujoco/half_cheetah/

[6]Hopper environment: https://www.gymlibrary.dev/environments/mujoco/hopper/

Table 2: Network architecture for DML-IL. For mixture of Gaussians output, we report the number of components. No dropout is used.

(a) Roll-out model $\hat{M}$

| Layer Type | Configuration |
|---|---|
| Input | state dim $\times$ 3 |
| FC + ReLU | Out: 256 |
| FC + ReLU | Out: 256 |
| MixtureGaussian | 5 components; Out: state dim $\times$ k |

(b) Policy model $\hat{\pi}_h$

| Layer Type | Configuration |
|---|---|
| Input | state dim$\times$ (k+3) |
| FC + ReLU | Out: 256 |
| FC + ReLU | Out: 256 |
| FC | Out: action dim |

$pos_{t-1} + \frac{pos_t - pos_{t-1}}{u_t^o}$ such that the imitator can infer information regarding $u_t^o$ from trajectory history.

# E  IMPLEMENTATION DETAILS

Experiments are carried out on a Linux server (Ubuntu 18.04.2) with two Intel Xeon Gold 6252 CPUs, and each experiment run uses a single NVIDIA GeForce RTX 2080 Ti GPU for neural network training.

## E.1  EXPERT TRAINING

The expert in the aeroplane ticket pricing environment is explicitly hand-crafted. For the Mujoco environments, we used the Stable-Baselines3 (Raffin et al., 2021) implementation of soft actor-critic (SAC) and the default hyperparameters for each task outlined by Stable-Baseline3. The expert policy is an MLP with two hidden layers of size 256 and ReLU activations, and we train the expert for $10^7$ steps.

## E.2  IMITATOR TRAINING

With the expert policy $\pi_E$, we generate 40 expert trajectories, each of 500 steps, following our previously defined environments. Specifically, the confounding noise is added to the state and actions and crucially $u_t^o$ is not recorded in the trajectories. The naive BC directly learns $\mathbb{E}[a_t \mid s_t]$ via supervised learning. ResiduIL mainly follows the implementation of Swamy et al. (2022b), where we adapt it to allow a longer confounding horizon $k > 1$. For DML-IL and BC-SEQ, a history-dependent policy is used, where we fixed the look-back length to be $k + 3$, where $k$ is the confounding horizon. BC-SEQ then just learns $\mathbb{E}[a_t \mid h_t]$ via supervised learning, and DML-IL is implemented with $K$-fold following Algorithm 2. The policy network architecture for BC, BC-SEQ, and ResiduIL are 2 layer MLPs with 256 hidden size. The policy network $\hat{\pi}_h$ and the mixture of Gaussians roll-out model $\hat{M}$ for DML-IL have a similar architecture, with details provided in Table 2. We use the AdamW optimizer with a weight decay of $10^{-4}$ and a learning rate of $10^{-4}$. The batch size is 64 and each model is trained for 150 epochs, which is sufficient for their convergence.

## E.3  IMITATOR EVALUATION

The trained imitator is then evaluated for 50 episodes, each 500 steps in the respective confounded environments. The average reward and the mean squared error between the imitator's action and the expert's action are recorded.

## F  PRACTICAL CONSIDERATIONS FOR DML-IL

DML-IL can also be implemented with $K$-fold cross-fitting, where the dataset is partitioned into $K$ folds, with each fold alternately used to train $\hat{\pi}_h$ and the remaining folds to train $\hat{M}$. This ensures unbiased estimation and improves the stability of training. The base IV algorithm DML-IV with $K$-fold cross-fitting is theoretically shown to converge at the rate of $O(N^{-1/2})$ (Shao et al., 2024), where $N$ is the sample size, under regularity conditions. DML-IL with $K$-fold cross-fitting (see Appendix G for details) will thus inherit this convergence rate guarantee.

**Discussion on the Confounding Noise Horizon.**    Note that Algorithm 1 requires the confounding noise horizon $k$ as input. Although the exact value of $k$ can be difficult to obtain in practice, any upper bound $\bar{k}$ of $k$ is sufficient to guarantee the correctness of Algorithm 1, since $h_{t-\bar{k}}$ is also a valid instrument. Ideally, we would like a data-driven approach to determine $k$. Unfortunately, the confounding horizon $k$, or equivalently the validity of $h_{t-k}$ as an IV, generally cannot be definitively verified using empirical data, especially the unconfounded instrument condition (i.e., $h_{t-k} \perp\!\!\!\perp u_t^\varepsilon$).

Therefore, we rely on the user to provide a sensible choice of $\bar{k}$ based on the environment that does not substantially overestimate $k$, informed by domain knowledge about the task. However, there exist tests that can indirectly check whether a candidate IV is valid, such as the overidentification tests (Hansen, 1982; Sargan, 1958), conditional independence tests between the instrument and the residual (Gretton et al., 2005; Fukumizu et al., 2008), and sensitivity analysis (Conley et al., 2012). It would be interesting future work to incorporate these methods to help identify $k$. In Appendix C.2, we additionally evaluate the performance and sensitivity of DML-IL under misspecification of $k$.

**Discussion on the Additive Noise Assumption.**    The additive noise assumption in Theorem 3.3 is a key identification assumption and is standard in IV regression (Pearl, 2000). If the additive noise is misspecified, e.g., multiplicative or complex non-linearity, then the derivation of the CMR in Equation (4) breaks down. However, this limitation of DML-IL arises from the fact that, with non-additive confounding noise and without further assumptions, the causal effect is generally unidentifiable (Imbens & Newey, 2009). Therefore, while the additive noise assumption may be simplistic in complex settings such as healthcare, it is the best we can do without further assumptions.

The validity of additive noise can often be justified through domain knowledge. For example, in physical systems such as drones or aircraft, directional environmental noises such as wind and vibrations affect the position of a drone or plane additively. In econometrics applications, confounding noises, when quantified in monetary terms, naturally aggregate additively into total cost or revenue. Finally, it is worth noting that this assumption only requires the expert action (i.e., the outcome) to have additive noise, whereas the relationship between the confounding noise and the state (i.e., the treatment) is unrestricted.

## G  BACKGROUND ON DML AND DML-IL WITH $K$-FOLD CROSS-FITTING

Double Machine Learning (DML) (Chernozhukov et al., 2018) is a statistical technique that debiases two-stage regressions. In the DML framework, a function of interest $f$ is estimated in two stages. In the first stage, some parameters (which can be infinite-dimensional functionals) that are necessary for the second stage estimation are estimated. In the second stage, first stage estimators are plugged in to estimate the function of interest $f$. Shao et al. (2024) utilised the DML framework to propose DML-IV, which is a two-stage IV regression algorithm. DML-IV is also a general CMR solver (see DML-CMR, a generalisation of DML-IV proposed by Shao et al. (2025)) that can be used to solve general CMR problems. In Shao et al. (2025), a score (criterion) function that describes general CMR problems was proposed; the score function guarantees Neyman orthogonality for estimating solutions to CMR problems. Our CMR objective $\mathbb{E}[a_t - \pi_h(h_t) \mid h_{t-k}] = 0$ fits directly into the CMR framework of DML-CMR. In our adaptation in Algorithm 1, the rollout model $\widehat{M}$ serves as the nuisance component, and the second stage estimates $\pi_h$ using this orthogonal score.

In Shao et al. (2025), the authors show that DML-CMR can achieve a $O(N^{-1/2})$ convergence rate, where $N$ is the sample size, if implemented with $K$-*fold cross-fitting* under some standard DML conditions. Next, we introduce Algorithm 2, which is a version of DML-IL with $K$-fold cross-

---

**Algorithm 2** DML-IL with $K$-fold cross-fitting

---

**Input:** Dataset $\mathcal{D}_E$ of expert demonstrations, Confounding noise horizon $k$, number of folds $K$ for cross-fitting
**Output:** A history-dependent imitator policy $\hat{\pi}_h$
Get a partition $(I_k)_{k=1}^K$ of dataset indices $[N]$ of trajectories
**for** $k = 1$ **to** $K$ **do**
    $I_k^c := [N] \setminus I_k$
    Initialize the roll-out model $\hat{M}_i$ as a mixture of Gaussians model
    **repeat**
        Sample $(h_t, a_t)$ from data $\{(\mathcal{D}_{E,i}) : i \in I_k^c\}$
        Fit the roll-out model $(h_t, a_t) \sim \hat{M}_i(h_{t-k})$ to maximize log likelihood
    **until** convergence
**end for**
Initialize the expert model $\hat{\pi}_h$ as a neural network
**repeat**
    **for** $k = 1$ **to** $K$ **do**
        Sample $h_{t-k}$ from $\{(\mathcal{D}_{E,i}) : i \in I_k\}$
        Generate $\hat{h}_t$ and $\hat{a}_t$ using the roll-out model $\hat{M}_i$
        Update $\hat{\pi}_h$ to minimise the loss $\ell := \|\hat{a}_t - \hat{\pi}_h(\hat{h}_t)\|_2$
    **end for**
**until** convergence

---

fitting, and discuss the specific conditions required for DML-IL to achieve $O(N^{-1/2})$ convergence rate.

## G.1 DML-IL with $K$-fold cross-fitting

Here, we outline DML-IL with $K$-fold cross-fitting. The algorithm is shown in Algorithm 2. The dataset is partitioned into $K$ folds based on the trajectory index. For each fold, we use the leave-out data, that is, indices $I_k^c := [N] \setminus I_k$, to train separate roll-out models $\hat{M}_i$ for $i \in [1..K]$. Then, to train a single expert model $\hat{\pi}_h$, we sample the trajectory history $h_{t-k}$ from each fold and use the roll-out model trained with the leave-out data to complete the trajectory and train $\hat{\pi}_h$. This technique is very important in Double Machine Learning (DML) literature (Shao et al., 2025; Chernozhukov et al., 2018) for it provides both empirical stability and $O(N^{-1/2})$ convergence rate guarantees.

The conditions required for this root-$N$ consistency are standard DML-CMR conditions ((Shao et al., 2025), Condition 4), which includes identifiability conditions, orthogonality and a nuisance convergence rate of $o(N^{-1/4})$. The identifiability conditions are satisfied if we have a valid instrument, and the orthogonality is guaranteed by the score function in Shao et al. (2025). The nuisance rate requires that our nuisance parameter converges at $\|\widehat{M} - M\|_2 = o(N^{-1/4})$, which is usually achieved by density estimation models such as mixture Gaussian (see discussion before Theorem 6 in Shao et al. (2025)). Therefore, DML-IL with $K$-fold cross-fitting will thus inherit this convergence rate guarantee if all the above conditions are satisfied.

