# OpenReview forum: "Causal Imitation Learning under Expert-Observable and Expert-Unobservable Confounding"
_ICLR.cc/2026/Conference — ICLR 2026 Poster_

### Official Review · Reviewer_2XU2 · 2025-10-29

**Soundness:** 2
**Presentation:** 2
**Contribution:** 3
**Rating:** 4
**Confidence:** 4

**Summary:**

This paper proposes a causal imitation learning (IL) framework that explicitly models two forms of hidden confounding. It leverages trajectory histories as instruments to convert confounded IL into a conditional moment restriction (CMR) problem and introduces DML-IL, an instrumental-variables–based algorithm to solve it.  The authors establish an upper bound on the imitation gap that recovers earlier results as special cases, and demonstrate in continuous-control benchmarks (including MuJoCo) that DML-IL consistently outperforms existing causal IL baselines.

**Strengths:**

- some important assumptions are clearly stated
- some motivating examples (e.g., Example 3.1) are provided
- an expansion of [3], with a longer confounding horizon $k > 1$

**Weaknesses:**

- Some important relevant papers are not included (e.g., [1]).
- Claims outpace evidence. The "unifying" scope is asserted more strongly than it is demonstrated.
- Some assumptions are hard to be verified in practice

Specific questions below.

**Questions:**

## Identifiability under missing expert covariates

- Your formulation assumes that the expert’s additional covariates $u_{t}^{o}$ are never directly observed in demonstrations (only indirectly via histories). Can you clarify whether your identification or bounds require any minimal observability or recoverability conditions on $u_{t}^{o}$?
- [1] studies the scenario where exact imitation is impossible. How does your analysis relate to partial-identification results showing impossibility from demonstrations when state/action/reward are jointly confounded [1]? You note that without assumptions the expert policy is unidentifiable; please delineate precisely which of your assumptions (e.g., Additive Noise, Confounding Noise Horizon) are sufficient to escape [1]’s impossibility and where residual partial identification remains. It might be better if the authors could add such discussions comparing with [1] in their paper.

## Confounding noise horizon

- Algorithm 1 requires k as input. However, in practice, how do we know k beforehand? Do we need to run causal discovery algorithms to obtain k? Additionally, can you provide a data-driven procedure used in your experiments, report sensitivity to $k$ or $\bar{k}$, and include overidentification or CI-based diagnostics per environment?
- In the worst case k=T−1, the instrument is the entire past. With continuous, high-dimensional states, both statistical and computational complexity can scale poorly. In that case, the scenario is similar to POMDP. Please characterize how sample complexity and runtime of DML-IL scale with k, T and state/action dimensionality.
- Since the validity $h_{t-k} \perp u_{t}^{\varepsilon}$ is very hard to be verified, how do the proof steps change if the instrument is slightly invalid (e.g., $\delta_{\mathrm{IV}}$ violation)? Robustness version of Theorem 4.5?
- Is $\delta$ treated as a structural constant, or can it be bounded/estimated from data (e.g., via proxy tests) ? Can you provide an example showing how large $\delta$ can be in typical MuJoCo settings?

## Additive-noise assumption

> Actions are generated via $a_{t}=\pi_{E}\left(s_{t}, u_{t}^{o}\right)+u_{t}^{\varepsilon}$.

-  What if in general like $a_{t}=\pi_{E}\left(s_{t}, u_{t}^{o}, u_{t}^{\varepsilon}\right)$ non-additive cases [1][2], how could you deal with them?
- Could you bound the bias of DML-IL under structured departures from additivity (e.g., small multiplicative terms) or provide a sensitivity analysis (e.g., "plausibly exogenous") approaches?

## Experiments

> we generate 40 expert trajectories, each of 500 steps, following our previously defined environments.

- Why 40×500 specifically? Please report sensitivity to the number/length of demos.

- K-fold details. You mention K-fold cross-fitting and give Algorithm 2. What K did you use in each experiment? Were folds stratified by environment/noise setting? How sensitive are results to K (1, 2, 5, 10)? Please include means ± CIs across seeds.

- Code and seeds. Please release code, random seeds, and environment configs (including the exact noise processes) so others can reproduce the roll-out model training and cross-fitting setup described in Appendix

## Scope

Finally, I strongly recommend the authors to change their claim about "**Unifying Framework**" for Causal Imitation Learning. There are lots of cases that cannot be covered by your framework (i.e., the current scope excludes several practically common regimes), e.g., violations of the additive-noise assumption (which is quite common), stochastic experts, sub-optimal experts, invalid history instruments, unknown or unbounded confounding horizons, partial observability, different causal diagrams and so on. These are important and active research fronts; acknowledging them explicitly—ideally with a short taxonomy of covered vs. uncovered regimes and a softened headline claim —would make the paper’s contributions more precise and credible while pointing concretely to valuable directions for future work.

I would be _inclined to raise my score_ if the revision clarifies scope as suggested and properly answers my questions above.

-----


[1] "Causal imitation for markov decision processes: A partial identification approach." Advances in neural information processing systems 37 (2024): 87592-87620.

[2] "Causal imitation learning via inverse reinforcement learning." The Eleventh International Conference on Learning Representations. 2023.

[3] "Causal imitation learning under temporally correlated noise." International Conference on Machine Learning. PMLR, 2022.

[4] "Invariant causal imitation learning for generalizable policies." Advances in Neural Information Processing Systems 34 (2021): 3952-3964.

---

> ### Author Response · Authors · 2025-11-25
>
> Dear Reviewer 2XU2,
>
> Thank you for taking the time to review our work. We appreciate your noting the clarity of the core assumptions, the usefulness of the motivating examples, and the generalization beyond the $k=1$ setting in prior work.
>
> **(W1) Additional related work:** Thank you for the pointer. We have added a thorough discussion of [1] to the related work section (cf. updated submission). We have also added a citation of [4] and we already cite [2] and [3]. Please let us know if we have missed any other important related work.
>
> **(W2) Clarifying the scope of the proposed framework:** We agree that our proposed framework does not cover certain other regimes, such as interactive IL and non-additive noise models. We believe that the main paper text is already clear about what settings are subsumed and which are not, e.g., we clearly name the settings covered by our framework and that we are considering non-interactive IL. Where it was not clear before, we made minor edits to clarify it in the revised version. We do agree that the title (“Unifying Framework”) suggests an even broader framework. In light of this, we’d be happy to change the title to “*Causal Imitation Learning under Expert-Observable and Expert-Unobservable Confounding*”. Let us know whether this title better suits the scope in your opinion, before we make the change officially on OpenReview. Please also see our **joint response** further above.
>
>
> **(W3) Assumptions hard to verify in practice:** Please see our responses to your detailed questions below.
>
>
> ---
>
>
> **(Q1) Identifiability under missing expert covariates:**
>
> 1. **Observability or recoverability conditions on $u_t^o$?**  Thank you for raising this point. In our setting, full identification of $\pi_E$ is only possible if $u_t^o$ is fully recoverable from the history $h_t$, otherwise $\pi_E$ cannot be point-identified. Our results do *not* require such recoverability. Instead, Theorem 4.5 expresses the imitation gap in terms of $\delta_{TV}(u_t^o, E_{\pi_E}[u_t^o \mid h_t])$, which captures how much information about $u_t^o$ the history reveals. The bound holds without additional observability assumptions, and improves as $h_t$ becomes more informative about $u_t^o$.
> 2. **Relation to partial-identification results in [1]:** The impossibility result in [1] stems from the fact that, under fully nonparametric hidden confounding of the state/action/reward dynamics, the underlying transition and reward functions are not identifiable from demonstrations. Without additional structure, there always exists a compatible MDP in which no policy is guaranteed to match expert performance. In our setting, the additive noise assumption and the finite confounding horizon *together* provide sufficient structure for the trajectory history to serve as a valid instrument and for the conditional moment in Eq. (4) to be identifiable. Under these assumptions we can identify the CMR target $\pi_h(h_t)=\mathbb{E}[\pi_E(s_t,u_t^o)\mid h_t]$, and Theorem 4.5 bounds its gap to the expert in terms of how much information about $u_t^o$ is present in the history. Full identification of $\pi_E$ still requires $u_t^o$ to be fully recoverable from $h_t$, which we do not assume (see the TV-distance term discussed above). We appreciate the pointer, and we have added [1] to our related works section and discuss the partial identifiability there.
>
>
> **(Q2) Confounding noise horizon:**
> 1. **Selecting $k$ in practice:** The confounding horizon $k$, or equivalently the validity of $h_{t−k}$ as an IV, generally cannot be definitively verified using empirical data, as discussed in Appendix F. Therefore, for DML-IL, we rely on the user to specify a reasonable upper bound on $k$, informed by domain knowledge about the task. However, there exist tests that can indirectly check whether a candidate IV is valid, such as the overidentification tests (Hansen, 1982; Sargan, 1958), conditional independence tests between the instrument and the residual (Gretton et al., 2005; Fukumizu et al., 2008), and sensitivity analysis (Conley et al., 2012). It would be interesting future work to incorporate these methods to help identify $k$.
> In our experiments, $k$ is known as our environments are simulations and we input the actual $k$ value into the algorithm. In Appendix C.2, we also provide a sensitivity analysis to the misspecification of $k$ for DML-IL. The results show in the HalfCheetah environment no significant degradation of performance when we use any $k \geq 6$ while the actual horizon is $k^*=10$, and the performance stays within 5% of the correctly specified case. However, for $k < 6$, we observed a steep degradation (performance is halved compared to the correctly specified case.)

---

> > ### Author Response · Authors · 2025-11-25
> >
> > **(Q2) Confounding noise horizon:** (continued)
> >
> > 2. **When $k=T-1$ and the ill-posedness:** We would like to first clarify that we use the history from $k$ steps ago as instruments, so when $k=T-1$, only the first state and action pair can be used as the instrument for the last state action pair (and not the entire past). When $k=1$, the entire history can be used as instruments.
> > When $k$ gets larger, it becomes more difficult to imitate the expert because the ill-posedness $\nu (\Pi,k)$ from Definition 4.2 increases according to Proposition 4.3, and the first term of the imitation gap bound derived in Theorem 4.5 scales linearly with $\nu(\Pi,k)$. This means that, to obtain the same imitation gap, a lower CMR error $\epsilon$ must be achieved by the algorithm which requires more samples. Therefore, **sample complexity** increases as $k$ increases. The dimension of the state and action space will affect the convergence of the CMR error $\epsilon$, since high-dimensional regression can lead to slow convergence and worse statistical stability. The **runtime** scales with the actual amount of computation required to train the neural network, so a higher input dimension (state and actions spaces) and a longer lookback of history input as the instrument will slightly increase the computation complexity.
> >
> > In our experiments, we set the lookback length of the imitator policy to $k+3$ as mentioned in Appendix E.2, so the runtime is relatively stable within the same environment when varying $k$. This choice ensures that the imitator has access to sufficient actual trajectory history from the dataset during training. We also provided a sensitivity analysis for different lookback lengths for the plane ticket environment, and did not observe noticeable differences in performance and the runtime, where each run took around 10 minutes. We report the average reward together with its standard deviation (in parentheses).
> >
> > | Lookback Length | k+3             | k+6             | k+12            |
> > | --------------- | --------------- | --------------- | --------------- |
> > | k=1             | 0.7990 (0.0397) | 0.7952 (0.0255) | 0.7936 (0.0195) |
> > | k=3             | 0.7053 (0.0268) | 0.6989 (0.0184) | 0.7027 (0.0232) |
> > | k=5             | 0.6452 (0.0131) | 0.6392 (0.0145) | 0.6389 (0.0106) |
> >
> >
> > 3. **Mild violation of confounding horizon assumption:** In **Appendix C.2**, the original submission already includes experimental results under mild violation of the confounding noise horizon assumption. To summarise, a mild violation typically induces small bias empirically, and once the violation rises above a certain threshold, IV regression begins to induce large bias. However, to the best of our knowledge, there is no theoretical framework that can analyse IV regression bias with respect to IV violation with guarantees. In fact, in a theoretical worst-case, a weak correlation between the IV and the hidden confounder could potentially cause the causal effect to be unidentifiable, rendering causal inference tools ineffective.
> >
> > 4. **Size of $\delta$:** Regarding $\delta_{TV}(u^o_t,\mathbb{E}_{\pi_E}[u^o_t| h_t])\leq\delta$, it very much depends on the environment and how $u^o_t$ interacts with the expert and the environment. In MuJoCo for example, $\delta$ depends entirely on how $u^o_t$ is introduced into the environment. For example, if $u^o_t$ is a target position that is not revealed in the trajectory history, $\delta$ would be 1 (note that the total variational distance is bounded between 0 and 1) and $\mathbb{E}[u^o_t| h_t]$ degenerates into a point distribution at $\mathbb{E}[u^o_t]$. However, if $u^o_t$ is a target speed and the environment is designed in a way that the agent can determine this target speed by looking at the rate of change in position in the history, $\delta$ can be as low as 0. Unfortunately, the actual value of $\delta$ is difficult to estimate in practice, since it involves an unobserved variable $u^o_t$.

---

> ### Author Response · Authors · 2025-11-25
>
> **(Q3) Additive noise assumption:**
>
> 1. **Extension to nonlinear confounding:** We first consider the most general model $Y=g(X,\epsilon(U))$, where the goal is to identify $E[Y|do(x)]$. Without further assumptions, causal identification is generally impossible. However, under additional independence and monotonicity assumptions between variables, causal identification may be feasible (Imbens and Newey, 2009; Trogovitsky, 2015). Alternatively, it is possible to upper and lower bound the causal effect (Balke and Pearl, 1994; Kilbertus et al., 2020), which is known as partial identification. However, applying partial identification tools can be challenging, as high-performing imitation often requires precise inference of expert actions, not potentially loose bounds.
> For more specific noise models, several causal identification methods exist, such as for IV-Poisson models with multiplicative error (Windmeijer, 2000), log-linear models (Terza et al., 2008), and ML-based approaches for multiplicative noise (Wu et al., 2022). These noise models are often motivated by specific applications. In principle, inspired by the above methods, DML-IL could be extended to handle non-additive noise, provided that the associated assumptions (e.g., functional form, independence conditions) are justified within the application context.
>
> 2. **Robustness under misspecification:** If the additive noise is misspecified, e.g., multiplicative or complex non-linearity, then the derivation of the CMR in Eq. (4) breaks which makes DML-IL unsound and ineffective. However, this limitation of DML-IL arises from the fact that, with non-additive confounding noise and without further assumptions, the causal effect is generally unidentifiable (Imbens and Newey, 2009). Therefore, while the additive noise assumption may be simplistic in complex settings, it is the best we can do without further assumptions. For more specific problems, methodologies with appropriate additional assumptions can be developed. Finally, it is worth noting that this assumption only requires the expert action to have additive noise, whereas the relationship between the confounding noise and the state is unrestricted.
>
>
>
> **(Q4) Experiments:**
>
> 1. 40x500 is a choice to get appropriate coverage of the state and action spaces. Empirically, we found that as long as we have a good number of samples (e.g., 40x500, 20x1000 and 40x1000), there is no noticeable performance difference. However, with many fewer samples (e.g., 20x500), performance degradation was observed.
>
> 2. **K-fold cross-fitting** is required to obtain independent estimates for the two stages (first stage to learn the roll-out model, second stage to learn \hat{$\pi_E$}) of the algorithm and empirically, a good choice of $K$ is between 5 and 10. In our experiments, we set $K=10$. A larger $K$ will in theory give better results because more data can be used to train the roll-out models, but it will be more computationally expensive since it will need to be trained $K$ times. Empirically, the results are not sensitive to the choice of $K$ at all as long as it is above 5.
>
> 3. **Code, the exact environment configuration and random seeds** are provided in the supplementary material.
>
> Please let us know if any concerns or questions remain.
>
> ####
>
> **Hansen, L. P. (1982).** Large sample properties of generalized method of moments estimators. Econometrica.
>
> **Sargan, J. D. (1958).** The Estimation of Economic Relationships using Instrumental Variables. Econometrica.
>
> **Gretton, A., Bousquet, O., Smola, A., & Schölkopf, B. (2005).** Measuring statistical dependence with Hilbert-Schmidt norms. Algorithmic Learning Theory.
>
> **Fukumizu, K., Gretton, A., Sun, X., & Schölkopf, B. (2008).** Kernel measures of conditional dependence. NeurIPS.
>
> **Conley, T. G., Hansen, C. B., & Rossi, P. E. (2012).** Plausibly exogenous. Review of Economics and Statistics.
>
> **Imbens, G. W., & Newey, W. K. (2009).** Identification and Estimation of Triangular Simultaneous Equations Models Without Additivity. Econometrica.
>
> **Torgovitsky, A. (2015).** Identification of Nonseparable Models Using Instruments With Small Support. Econometrica.
>
> **Balke, A., & Pearl, J. (1994).** Counterfactual Probabilities: Computational Methods, Bounds and Applications. UAI.
>
> **Kilbertus, N., Kusner, M., & Silva, R. (2020).** A Class of Algorithms for General Instrumental Variable Models. NeurIPS.
>
> **Windmeijer, F. (2000).** Moment conditions for fixed effects models with endogenous regressors. Economics Letters.

---

> ### Comment · Reviewer_2XU2 · 2025-11-25
>
> The authors have satisfactorily addressed my concerns, particularly regarding the related work and the scope of the framework. I appreciate the decision to update the title and clarify the limitations in Section 6.
>
> In my view, generalizing Swamy et al. to settings where $k>1$, backed by some of theoretical guarantees, is a valuable contribution to the causal imitation learning and broader RL/causality communities. Consequently, I am increasing my score. Good luck.

---

### Official Review · Reviewer_mgqr · 2025-10-31

**Soundness:** 3
**Presentation:** 3
**Contribution:** 2
**Rating:** 6
**Confidence:** 3

**Summary:**

This paper proposes a unifying framework for causal imitation learning in the presence of hidden confounders, distinguishing between two types: variables observed by the expert but not the imitator (expert-observable), and noise hidden from both parties (expert-unobservable). By leveraging trajectory histories as instrumental variables and assuming a bounded confounding noise horizon along with additive noise structure, the authors reformulate the imitation problem as a Conditional Moment Restriction (CMR). They introduce DML-IL, a novel algorithm based on double machine learning to solve this CMR, and provide a theoretical upper bound on the imitation gap that generalizes prior results. Empirical evaluations on both a custom airline pricing task and modified MuJoCo environments demonstrate that DML-IL outperforms existing causal imitation baselines when both types of confounders are present.

**Strengths:**

The paper is clearly structured and presents a compelling case for a more holistic approach to addressing hidden confounding in imitation learning. The proposed framework elegantly subsumes several prior settings, and the experimental results convincingly validate the method’s effectiveness across diverse environments. The theoretical analysis is rigorous and meaningfully connects to established results in causal inference and econometrics.

**Weaknesses:**

- The framework relies on two key assumptions that may limit its applicability in certain domains. First, the additive noise assumption (i.e., that unobservable confounders affect actions additively) is essential for identifiability but may be overly restrictive in complex systems where interactions are nonlinear or multiplicative (e.g., biological or financial systems). While this assumption is standard in instrumental variable literature, its validity must be carefully assessed per application.

- The method requires knowledge (or a valid upper bound) of the confounding noise horizon k —the temporal range over which unobservable noises remain correlated. Although the paper discusses practical heuristics and conditional independence tests to estimate k, this parameter is not identifiable from data alone, and misspecification (especially severe underestimation) can degrade performance, as shown in the ablation studies.

**Questions:**

- How robust is the proposed method against model misspecification, particularly with respect to the noise horizon at Assumption 3.2?

---

> ### Author Response · Authors · 2025-11-25
>
> Dear Reviewer mgqr,
>
> Thank you for reviewing our work and for your thoughtful feedback. We appreciate your positive comments on the overall structure of the paper, the breadth of the framework, the empirical validation, and the connection to established theory.
>
> **(W1, W2) Assuming additive noise and confounding noise horizon:** We agree that the additive-noise assumption and the need for an upper bound on the confounding horizon $k$ are important modeling choices whose applicability depends on the domain. We address both issues in the joint response further above.
>
> **(Q1) Robustness against misspecification:** We have included additional results on the robustness of the proposed method to underestimating the noise horizon in Table 1 in Appendix C.2, where we also more generally discuss misspecification. To summarise, a mild misspecification typically induces small bias. For example, the results show in the HalfCheetah environment no significant degradation of performance when we use any $k \geq 6$ while actual horizon is $k^*=10$, and the performance stays within 5% of the correctly specified case. However, for $k < 6$, we observed a steep degradation (performance is halved compared to the correctly specified case.)
>
> Please let us know if you have any additional questions or concerns.

---

### Official Review · Reviewer_YSLm · 2025-10-31

**Soundness:** 3
**Presentation:** 3
**Contribution:** 3
**Rating:** 6
**Confidence:** 3

**Summary:**

This paper proposes a unifying framework for causal imitation learning (IL) with hidden confounders, distinguishing between expert-observable confounders (variables observed by the expert but not the imitator) and expert-unobservable confounders (confounding noise hidden from both). By leveraging trajectory histories as instrumental variables, the authors reformulate causal IL as a Conditional Moment Restriction (CMR) problem and propose DML-IL, an algorithm that solves this CMR via instrumental variable regression. They provide an upper bound on the imitation gap and demonstrate empirically on continuous state-action environments that DML-IL outperforms existing causal IL baselines.

**Strengths:**

- Originality: The distinction between expert-observable and expert-unobservable confounders provides a novel and more realistic problem formulation that generalizes several prior works. The neat reframing of causal IL into instrumental variables and CMR problems is elegant and theoretically well-motivated.
- Quality: The theoretical analysis is solid, with the neat theory on the imitation gap bound (Theorem 4.5) that recovers prior results as special cases (Corollaries 4.6 and 4.7).
- Clarity: The paper is generally well-written with clear motivation through Example 3.1 and good use of figures (especially Figure 1). The connection between hidden confounders and the CMR formulation is explained systematically.
- Significance: The unifying framework addresses a realistic scenario where multiple confounding factors coexist, which is practically important and has been overlooked in prior work that typically addresses confounders in isolation.

**Weaknesses:**

- Limited scope of unification: The framework doesn't truly subsume Vuorio et al. (2022), which solves the problem through environment interaction. The experimental results don't include comparisons to interactive scenarios, limiting the claim of being a "unifying" framework. The paper focuses exclusively on offline IL from fixed demonstrations.
- Gap between theory and experiments: No analysis is provided comparing experimental results to the theoretical bounds from Theorem 4.5, making it unclear how tight these bounds are in practice.
- Algorithm derivation unclear: The connection between the CMR error minimization and Algorithm 1's two-stage procedure needs better motivation. The paragraph below Algorithm 1 explaining why training on expert trajectories does not work, is difficult to follow.
- Restrictive assumptions: The framework requires knowledge of (or an upper bound on) the confounding noise horizon k, which cannot be verified empirically. The additive noise assumption (Assumption 3.3) is quite strong and limits applicability. The authors don't fully discuss the effect of these limitations.

Minor:
- In Definition 4.2, the text describes "the ratio between the learning error... and its L2 error" but the equation shows the reciprocal of this ratio (L2 error in numerator, learning error in denominator).

**Questions:**

- How do the experimental results quantitatively compare to the theoretical bounds from Theorem 4.5? Can the authors provide concrete values for the variables in the theorem in the experimental settings to assess bound tightness?

---

> ### Author Response · Authors · 2025-11-25
>
> Dear Reviewer YSLm,
>
> Thank you for your time and effort in reviewing our submission and your constructive feedback. We are glad to hear that you found the proposed framework original and significant, the theoretical analysis neat, and the exposition clear and well-motivated.
>
> **(W1) Clarifying scope of unification:** Please see our joint response above.
>
> **(W2, Q1) Tightness of theoretical bounds and comparison to experimental results:** Our theoretical result provides an upper bound on the imitation gap for the worst-case scenario, where the confounding noise $u^\epsilon_t$ behaves adversarially at each step, and the trajectory history reveals minimal information about $u^o_t$. This bound is intentionally conservative and not representative of the average-case performance. Empirically, the actual imitation gap is likely to be smaller.
> Unfortunately, the exact bounds are challenging to compute because the ill-posedness of the CMR problem, which is a key component of the bound, is often intractable to estimate in practice. Nevertheless, the theoretical analysis remains valuable: it provides worst-case guarantees and establishes that, as the CMR optimisation error decreases, the imitation gap should also shrink, providing asymptotic guarantees on the performance.
>
> **(W3) Clarifying derivation of Algorithm 1 and its link to CMR error minimization:**
> Algorithm 1 is a direct two-stage implementation of the CMR condition
> $\mathbb{E}[\,a_t - \pi_h(h_t)\mid h_{t-k}\,] = 0$. To estimate $\pi_h$, we need to evaluate both conditional expectations inside this moment. This motivates the two stages. **First stage (rollout model):** We learn a model $\hat M$ that approximates the conditional distribution $F(\bar h_t \mid h_{t-k})$. This lets us estimate $\mathbb{E}[a_t \mid h_{t-k}]$ via $\hat a_t = \hat M(h_{t-k})$, and it also lets us simulate $\hat h_t \sim F(\cdot\mid h_{t-k})$ for use in the second-stage regression.
> **Second stage (CMR regression):** We then solve $\min_{\pi_h}|\hat a_t - \pi_h(\hat h_t)|2^2$, which is the empirical version of minimizing the CMR error $|\mathbb{E}[a_t\mid h{t-k}] - \mathbb{E}[\pi_h(h_t)\mid h_{t-k}]|$. Here, $\hat a_t$ estimates $\mathbb{E}[a_t\mid h_{t-k}]$ and the sample mean of $\pi_h(\hat h_t)$ estimates $\mathbb{E}[\pi_h(h_t)\mid h_{t-k}]$.
>
> **Why rollouts instead of using the expert trajectory directly?** The expert’s future trajectory after $h_{t-k}$ is confounded with the current state and action through the unobserved noise $u_t^\epsilon$, so it does not represent draws from the conditional distribution of future histories given $h_{t-k}$. Rolling out from $h_{t-k}$ with $\hat M$ removes this confounding and provides the correct conditional distribution required by the CMR moment. For each $t$, we use the real expert history up to time $t-k$ and only simulate the future segment that must be conditionally independent of $u_t^\epsilon$. We have added a short explanation of this point in the revised paper, and refer to Section 3 of Shao et al. (2024) for the general CMR estimator that Algorithm 1 instantiates.
>
> **(W4) Restrictive assumptions (confounding horizon and additive noise):** Please see our joint response above.
> **Minor: Typo** Apologies, we have changed the text to “the ratio between the $L_2$ error of the imitator to the expert policy, and the learning error of the imitator following our CMR objective.”
>
> Please let us know if you have any additional questions or concerns.
>
> **Daqian Shao et al., (2024).** Learning Decision Policies with Instrumental Variables through Double Machine Learning, Proceedings of ICML 2024.

---

### Official Review · Reviewer_iKXV · 2025-11-01

**Soundness:** 2
**Presentation:** 3
**Contribution:** 2
**Rating:** 2
**Confidence:** 3

**Summary:**

This paper proposes a framework for causal imitation learning with hidden confounders that distinguishes two types of latent confounders: (i) expert-observable confounders (visible to the expert but not to the imitator) and (ii) expert-unobservable confounders (hidden to both). Using trajectory histories as instrumental variables, the approach reduces imitation learning to a conditional moment restriction problem. They prove an imitation-gap upper bound that recovers prior bounds as special cases, and provide empirical validation on a toy pricing environment and modified MuJoCo tasks where their algorithm outperforms some baselines.

**Strengths:**

- Casting imitation under confounding as a CMR with trajectories as instruments is conceptually simple and lets the authors leverage existing IV/CMR machinery.

- The paper shows corollaries recovering Swamy et al. results as special cases.

**Weaknesses:**

A confusing point of this paper is the arrow from $u_\epsilon$ to $a$, which means that even the expert does not have full control over its actions--the expert might choose to take one action, but another action take place in the end because of $u_\epsilon$. I did not get how this is justified.

---
>"We assume the confounding noise is additive to the action, which is standard in causal inference. Without this assumption, the causal effect becomes unidentifiable."

I do not think that this is really 'standard'. The claim of unidenitifiability in case it is not assumed also requires more elaboration, as the setting of this paper is not exactly that of Balke and Pearl 1994. Most importantly, this assumption, as well as the first equation on page 5 at the beginning of Section 4 only make sense if the action $a_t$ is continuous, which is a point that needs be clarified.

---
It is not clear how Assumption 3.2 implies what is being used in the derivations afterwards. There is a missing step in the proofs which doesn't sound straightforward to me. Unless I'm missing something, there is an implicit assumption made that is not mentioned in the paper. See my question 2 below.

---
Algorithm 1 is adapted from Shao et al, 2024, but with the current form of adaptation, it is not clear whether or not convergence guarantees transfer from that paper. See my question 1.

---
There are a few symbols and notions that are used on the go without being defined at all, which makes following the paper difficult. For example, policy $\pi_h$ is defined as a function from $\mathcal{H}$ to $\Delta(A)$, but $\Delta()$ is never defined. So it is not even clear what we are looking for.
Same goes for dim(A) in Corollary 4.7 which is never defined, although it is guessable unlike $\Delta(A)$.

---
The plots of Figures 2 and 3 are too small, and the font size is so small that the text is almost not readable.

---
Minor:

Figure 1: dotted -> dashed, last paragraph of page 9: one limitation are -> one limitation is

**Questions:**

1. Could you please clarify why algorithm 1 is a double machine learning algorithm? What are the nuisance parameters? How is the Neyman orthogonality guaranteed? It is simply mentioned that under "regularity assumptions" the estimator converges at root-n rate. I do not think achieving root-n convergence (especially without cross-fitting) is as simple as that, perhaps Donsker-type conditions are meant as "regularity conditions?" I believe the authors should at least name the assumptions for convergence, and with the Gaussian mixture roll-out, a more careful treatment is required.

---
2. When proving Eq. (4), under the derivation of CMR problem on page 5, the authors use $u^\epsilon_t \perp\\!\\!\\!\perp h_{t-k}$, whereas the assumption they have made is $u^\epsilon_t \perp\\!\\!\\!\perp u^\epsilon_{t-k}$. Could you elaborate on how this assumption implies $u^\epsilon_t \perp\\!\\!\\!\perp h_{t-k}$? It does not sound trivial to me.

---
3. Example 3.1 is meant to make the paper clearer, but it sort of adds to the confusion. What is the action set here? Does the agent set the price or does it set the profit margin? If it sets the price, then the example is confusing and does not fit into the paper. If it sets the profit margin, then I find the example unrealistic as this would mean that the airline can set the profit margin but not the price?

---

> ### Author Response · Authors · 2025-11-25
>
> Dear Reviewer iKXV,
>
> Thank you for taking the time to review our work. We appreciate your recognition of the simplicity of our CMR formulation with trajectory-based instruments and of the fact that our results recover prior work as special cases.
>
> **(W1) Clarifying arrow from $u^\epsilon_t$ to $a_t$:** In our model, the expert computes the action prescribed by its policy $\pi_E (s_t, u_t^o)$, but the *realised* action recorded in the dataset is $a_t = \pi_E (s_t, u_t^o) + u_t^\epsilon$. The arrow
> $u_t^\epsilon \to a_t$ expresses that the executed action is perturbed by the unobserved environmental noise (e.g., actuation noise, exogenous cost shocks, logging noise). This perturbation is *unobserved* by the expert, which is why it is not an argument of $\pi_E$. This follows the standard IV/SEM formulation in which the expert selects an action based on its information, and the environment adds an unobserved disturbance before the action takes effect in the transition model. Thus, the arrow does *not* mean that the expert “loses control”, it simply makes explicit that the learner observes a confounded realised action $a_t$. This problem setting is also motivated and considered by Swamy et al. (2022).
>
> **(W2) Additive noise assumption & identifiability:**
> If we drop the additive structure and allow the realised action to follow an unrestricted nonparametric equation $a_t = g(s_t, u_t^o, u_t^\epsilon)$, then the contributions of $\pi_E (s_t, u_t^o)$ and $u_t^\epsilon$ cannot be separated from observational data.  This places us in the same setting as the first equation in Section 3.1 of Balke and Pearl (1994) and other classic causal inference literature (e.g., Imbens and Newey, 2009), where its shown that without further assumptions, the causal effect is unidentifiable and we can only derive bounds (which may be vacuous) on the causal effect.
>
> **Continuous vs. discrete action spaces:** We agree that the paper should explicitly discuss the assumption on the action space, and we have added this clarification to Section 4.1 in the revised submission (highlighted in colour). To briefly recap the added discussion, the target quantity $\pi_h(h_t)=E[\pi_E(s_t,u_t^o)|h_t]$ is well-defined for both continuous and discrete $a_t$, because the mathematics and derivation of Eq. (4) work for both. However, in the algorithm and the theoretical analysis of the imitation gap, we implicitly assumed that $a_t$ is continuous such that $\pi_h(h_t)$ is an allowed action by the imitator. In practice, if the action space is discrete, we will need a mapping that maps $\pi_h(h_t)$ to to the action space, such as treating $\pi_h(h_t)$ as the logits output to the action space.
>
>
> **(W3, Q2) Derivation of conditional independence:** Regarding the derivation of $h_{t-k}\perp u^\epsilon_t$ used for Eq. (4), this follows from standard d-separation rules for causal graphs (Pearl et al., 2016). To establish this, we must verify that all paths from $h_{t-k}=(s_1,a_1,...,s_{t-k})$ to $u^\epsilon_t$ are blocked in the graph, meaning that $h_{t-k}$ is d-separated from $u^\epsilon_t$, which implies $h_{t-k} \perp u^\epsilon_t$. From our causal graph in Figure 1, we see that any paths from $h_{t-k}$ to $u^\epsilon_t$ must pass through a collider structure, specially through either $s_t\rightarrow a_t\leftarrow u^\epsilon_t$ or $a_t\rightarrow s_{t+1}\leftarrow u^\epsilon_t$. Furthermore, potential paths through hidden confounders are ruled out because there are no direct causal paths between $u^\epsilon_{t-k}$ and $u^\epsilon_t$, as required by Assumption 3.2. Thus, all paths from $h_{t-k}$ to $u^\epsilon_t$ are blocked by d-separation, and we can conclude that $h_{t-k}\perp u^\epsilon_t$. We have added a full proof in Appendix B.1 to verify the IV conditions for $h_{t-k}$.

---

> > ### Author Response · Authors · 2025-11-25
> >
> > **(W4, Q1) Double machine learning (DML) and convergence rate:** Our CMR objective $\mathbb{E}[a_t - \pi_h(h_t)\mid h_{t-k}]=0$ fits directly into the linear CMR framework of Shao et al. (2024). Their DML-CMR score is Neyman-orthogonal by construction (Eq. (6) in that paper), and in our adaptation the rollout model $\hat M$ serves as the nuisance component, exactly analogously to the first-stage regression in DML-IV. The second stage estimates $\pi_h$ using this orthogonal score, which is why Algorithm 1 follows the DML structure.
> > We do not claim root-$n$ convergence for Algorithm 1. As noted on line 292 and detailed in Appendix G, root-$n$ requires $K$-fold cross-fitting, and Algorithm 2 provides this cross-fitted version. The assumptions needed for root-$n$ consistency are the standard DML-CMR conditions (identification, orthogonality, and a nuisance rate such as $||\hat M - M||_2 = o_p(N^{-1/4})$), matching Condition 4 in Shao et al. (2024). Mixture-model rollouts often satisfy this slow-rate requirement (Ho et al., 2022).
> > Algorithm 1 is shown without cross-fitting for clarity of presentation, and any CMR solver (e.g., DFIV or DeepGMM; see Appendix C.1) can be substituted, and the theoretical DML guarantees are not the central contribution of our paper. We have added a short clarification of the nuisance component and orthogonality in Appendix G.
> >
> > **(W5, W6, W7) Missing definitions and typos:** We apologise for the confusion caused by notation. We use the common convention where $\Delta(\mathcal{A})$ denotes the set of probability measures over $\mathcal{A}$ and $dim(\mathcal{A})$ is the dimension of the space $\mathcal{A}$. We clarified these notations in the revised version. We have also fixed the other formatting issues and typos in the paper.
> >
> >
> > Please let us know if any questions or concerns remain.
> >
> >
> > ####
> >
> > **Swamy et al., (2022).** Causal imitation learning under temporally correlated noise, ICML 2022.
> >
> > **Balke, A., & Pearl, J. (1994).** Counterfactual Probabilities: Computational Methods, Bounds and Applications. UAI.
> >
> > **Imbens, G. W., & Newey, W. K. (2009).** Identification and Estimation of Triangular Simultaneous Equations Models Without Additivity. Econometrica.
> >
> > **Pearl, J., Glymour, M., & Jewell, N. P. (2016).** Causal Inference in Statistics: A Primer. John Wiley & Sons.
> >
> > **Daqian Shao et al., (2024).** Learning Decision Policies with Instrumental Variables through Double Machine Learning, Proceedings of ICML 2024.
> >
> > **Nhat Ho et al., (2022).** Convergence Rates for Gaussian Mixtures of Experts. JMLR.

---

### Author Response · Authors · 2025-11-25
**Joint Response**

Dear Reviewers,

We'd like to jointly address common questions about the assumptions we make and the scope of the unifying framework below. Answers to specific questions can be found in the individual responses further below.

Several comments focused on the **two main assumptions** in our framework: (i) additive confounding noise, and (ii) knowing (or having an upper bound on) the confounding noise horizon $k$. Without them, the IV conditions do not hold and we cannot guarantee the correctness of the moment equation or the imitation gap bound. We do not claim that these assumptions hold universally. We provided in the paper some examples where these assumptions are known to hold (e.g., drone and ticket sales), while we acknowledge that our method is not applicable to all scenarios, e.g., in cases where non-linear confounding is typical. We discuss conditional independence tests and other practical diagnostics that can help practitioners to test for the satisfaction of these assumptions or the validity of instrumental variables. Please also refer to **Appendix F**, where we thoroughly discuss the “Practical Considerations for DML-IL”.

Regarding the **scope of this paper**, we agree that the claim of a unifying framework may be too strong. While our proposed confounded MDP framework generalised several prior works in confounded IL as discussed in the main text and in Appendix A, our algorithm DML-IL relies on the aforementioned assumptions and focuses on IL from a fixed set of confounded expert demonstrations without further interactions.

There are many active research fronts that consider causal identification with non-additive noise, partially observable covariates and invalid instruments. They are beyond the scope of this paper and are orthogonal to our work. It would be an interesting research direction though to consider our confounded MDP framework in these problem settings.

Following the suggestion to clarify the scope, we have changed the title to "**Causal Imitation Learning under Expert-Observable and Expert-Unobservable Confounding**" in order to more accurately capture the contribution and scoping of our work, and have expanded the discussion in Section 6 to clarify the limitations and scoping. We will change the title on OpenReview as well once we've heard your feedback on this.

---

### Author Response · Authors · 2025-12-03
**Summary of Rebuttal**

Dear **New Area Chair**,

We understand that due to the recent incident, you have been newly assigned to our submission. We here summarise the reviewers’ main concerns and our resolutions for your convenience.

**Reviewer iKXV (Score 2):**
- **Initial Concerns:** The reviewer expressed confusion regarding the problem formulation, specifically the arrow $u_{\varepsilon} \to a$ (interpreting it as the expert losing control), and the derivation of Eq. (4). They also questioned the "standard" nature of the additive noise assumption, the identifiability of the setting. All other concerns were minor, e.g., notation, which we carefully addressed in the revised submission.
- **Resolution:**
    - **Clarified Formulation:** We explained that the arrow $u_{\varepsilon} \to a$ represents standard environmental noise (e.g., actuation noise) that is *unoberserved* by the expert and perturbs the realized actions — a formulation consistent with established work like Swamy et al. (2022).
    - **Identifiability:** We clarified that without the additive noise assumption (e.g., allowing arbitrary $a_t = g(s_t, u_t^o, u_t^\epsilon)$), our problem becomes unidentifiable as established in classic causal inference literature (e.g., Balke & Pearl, 1994), making identifying the causal effect impossible without further constraints.
    - **Derivation of Eq. (4):** To fully clarify the independence relationship $h_{t-k}\perp u^\epsilon_t$ we used for deriving Eq. (4), we provide an explicit proof for this statement, and that $h_{t-k}$ is a valid instrument, in Appendix B.1.

Unfortunately, the reviewer did not respond before the incident. We believe that their main concerns stem from **misunderstanding** the causal IL setting with expert-unobservable confounders, which we have fully clarified in the rebuttal and in the paper.

**Reviewer YSLm (Score 6):**
- **Initial Concerns:** The reviewer felt the "unifying" framing was too broad. They also noted a gap between the theoretical worst-case bounds and experimental results and requested clarification on the derivation of Algorithm 1.
- **Resolution:**
    - **Title & Scope:** Following the reviewers' suggestion, we changed the title to "Causal Imitation Learning under Expert-Observable and Expert-Unobservable Confounding" and explicitly clarified that our unifying framework applies to non-interactive settings in the revised paper.
    - **Theory vs. Experiments:** We clarified that Theorem 4.5 provides conservative worst-case bounds on the imitation gap, whereas experiments reflect average-case performance; the theory establishes that minimizing CMR error asymptotically reduces the imitation gap.
    - **Algorithm 1:** We clarified that the "rollout" step is necessary because future expert trajectories are confounded; the rollout model $\hat{M}$ generates the correct conditional distribution required for the CMR moment.

Unfortunately, the reviewer did not get to respond before the discussion period was prematurely terminated. We believe that our rebuttal and particularly the clarification of the scope in our revised submission have **fully addressed** the reviewer’s concerns while the reviewer already highlighted the **originality**, **quality**, **clarity** and **significance** of our work in their review.

**Reviewer mgqr (Score 6):**
- **Initial Concerns:** The reviewer found the assumptions of additive noise and knowing the confounding horizon $k$ (or an upper bound) to be restrictive. They asked about robustness to model misspecification of the confounding horizon.
- **Resolution:** We acknowledged that while these assumptions are necessary for our IV conditions and bounds to hold, they are not universal; we discuss diagnostics (e.g., conditional independence tests in Appendix F) to help practitioners verify them. We also pointed to the experiments (Appendix C.2) showing that the method is robust to mild misspecification of the confounding horizon.

**Reviewer 2XU2 (Score 4 -> 8):**
- **Initial Concerns:** The reviewer originally raised concerns about missing related work, the "unifying" scope claim, and that assumptions are hard to verify in practice.
- **Resolution:**
    - **Related Work:** We incorporated the suggested related work (e.g., partial identification approaches).
    - **Scope:** We clarified the scope and following the reviewers’ comments changed the title to "Causal Imitation Learning under Expert-Observable and Expert-Unobservable Confounding” and clarified the scope in the revised paper.
    - **Assumptions:** While these assumptions cannot be fully verified in practice, we discuss practical considerations and diagnostics to detect violations in Appendix F and show empirically in Appendix C.2 that DML-IL remains stable under mild misspecification of the confounding horizon.

The reviewer revised their opinion of the paper well before the incident and recommended acceptance (as poster) as their concerns were resolved by our rebuttal and revisions.

---

### Meta-Review · Area_Chair_DJtY · 2026-01-07

**Summary:**

Overall, reviewers agree the paper targets an important and realistic version of causal imitation learning by explicitly separating expert-observable and expert-unobservable confounding, and by casting the problem into a trajectory-IV / CMR form with a clean imitation-gap analysis. The main concerns were: (i) whether the “unifying” claim overreaches the actual scope (offline-only, additive noise, known/upper-bounded confounding horizon), (ii) whether the key assumptions and IV validity are sufficiently motivated, clearly stated, and practically actionable, and (iii) clarity/rigor issues around the moment derivation, algorithmic instantiation of DML/CMR, and missing related work.

The rebuttal and revision address these points in a satisfactory way: the scope is tightened (including a title change), related work is expanded (notably partial-identification perspectives), and the main technical misunderstandings are clarified with added proofs/derivations and practical diagnostics/sensitivity analyses for horizon misspecification. With these clarifications, the remaining limitations read as explicit scope constraints. Given this, I lean accept.

**Reviewer Concerns:**

Concerns addressed by the rebuttal.

Formulation and causal interpretation (iKXV).
The rebuttal clarifies the causal graph, the meaning of the arrow from policy to realized action (as execution/logging noise), and adds a formal proof of the conditional independence used for Eq. (4). This resolves the main soundness and confusion issues.

Additive noise and identifiability (iKXV, mgqr, 2XU2).
The authors clarify that additivity and a bounded confounding horizon are structural assumptions needed for identification, not universal claims. They add discussion, references, and sensitivity experiments for horizon misspecification.

Scope and “unifying” claim (YSLm, 2XU2).
The scope is narrowed and stated explicitly (offline, non-interactive, specific confounding structure), including a title change and a limitations section.

Relation to prior work (iKXV, 2XU2).
Missing related work is added and the contribution is repositioned as extending and combining existing lines rather than replacing them.

Algorithmic clarity and DML structure (iKXV, YSLm).
The role of CMR, the two-stage procedure, the rollout model as nuisance, and the need for cross-fitting are clarified.

Robustness and reproducibility (mgqr, 2XU2).
Sensitivity to horizon misspecification, data size, and folds is reported, and code/configuration details are clarified.

Concerns that remain partially outstanding.

Restrictive assumptions.
Additive noise and a bounded confounding horizon are still strong and limit applicability, especially in complex real-world domains.

Limited theoretical coverage.
The framework does not handle non-additive noise, invalid instruments, or interactive settings, and provides no guarantees there.

Overall, the rebuttal resolves the main technical, clarity, and scope concerns. The remaining issues are limitations on generality rather than blockers for acceptance.

**Reviewer Scores:**

See above.

---

### Decision · Program_Chairs · 2026-01-26

Accept (Poster)